# Context-aware single-cell multiomics approach identifies cell-type-specific lung cancer susceptibility genes

Erping Long [1,8,9], Jinhu Yin [1,9], Ju Hye Shin[2,9], Yuyan Li[3,9], Bolun Li[1], Alexander Kane[1], Harsh Patel [1], Xinti Sun [3], Cong Wang[3], Thong Luong[1], Jun Xia[4], Younghun Han[5], Jinyoung Byun [5], Tongwu Zhang [1], Wei Zhao [1], Maria Teresa Landi [1], Nathaniel Rothman[1], Qing Lan [1], Yoon Soo Chang [2], Fulong Yu[6], Christopher I. Amos [5], Jianxin Shi [1], Jin Gu Lee[7,10] ✉, Eun Young Kim[2,10] ✉ & Jiyeon Choi [1,10] ✉

Genome-wide association studies (GWAS) identified over fifty loci associated with lung cancer risk. However, underlying mechanisms and target genes are largely unknown, as most risk-associated variants might regulate gene expression in a context-specific manner. Here, we generate a barcode-shared transcriptome and chromatin accessibility map of 117,911 human lung cells from age/sex-matched ever- and never-smokers to profile context-specific gene regulation. Identified candidate *cis*-regulatory elements (cCREs) are largely cell type-specific, with 37% detected in one cell type. Colocalization of lung cancer candidate causal variants (CCVs) with these cCREs combined with transcription factor footprinting prioritize the variants for 68% of the GWAS loci. CCV-colocalization and trait relevance score indicate that epithelial and immune cell categories, including rare cell types, contribute to lung cancer susceptibility the most. A multi-level cCRE-gene linking system identifies candidate susceptibility genes from 57% of the loci, where most loci display cell-category-specific target genes, suggesting context-specific susceptibility gene function.

Lung cancer is the leading cause of cancer deaths and affects diverse populations worldwide[1]. Although the major risk factor for lung cancer is tobacco smoking, up to 25% of lung cancers arise in never-smokers[2]. Lung cancer heritability is estimated as ~8–20%[3–6], and genome-wide association studies (GWAS) have identified over 50 risk loci so far.

Notably, a substantial proportion of these loci are specifically observed in subgroups based on histological subtypes, ancestry, and smoking status[7–9], suggesting heterogeneous genetic mechanisms of lung cancer susceptibility. While several biological pathways (e.g., telomere biology, immune response, DNA damage repair) have been highlighted

[1]Division of Cancer Epidemiology and Genetics, National Cancer Institute, Bethesda, MD, USA. [2]Department of Internal Medicine, Yonsei University College of Medicine, Seoul, Republic of Korea. [3]Institute of Basic Medical Sciences, Chinese Academy of Medical Sciences and Peking Union Medical College, Beijing, China. [4]Department of Biomedical Sciences, Creighton University, Omaha, NE, USA. [5]Institute for Clinical and Translational Research, Baylor College of Medicine, Houston, TX, USA. [6]Guangzhou National Laboratory, Guangzhou International Bio Island, Guangzhou, China. [7]Department of Thoracic and Cardiovascular Surgery, Yonsei University College of Medicine, Seoul, Republic of Korea. [8]Present address: Institute of Basic Medical Sciences, Chinese Academy of Medical Sciences and Peking Union Medical College, Beijing, China. [9]These authors contributed equally: Erping Long, Jinhu Yin, Ju Hye Shin, Yuyan Li. [10]These authors jointly supervised this work: Jin Gu Lee, Eun Young Kim, Jiyeon Choi. ✉e-mail: CSJGLEE@yuhs.ac; narae97@yuhs.ac; jiyeon.choi2@nih.gov

from these GWAS loci by single-locus-based studies[8], functional variants, and susceptibility genes are uncharacterized from most lung cancer GWAS loci.

Identifying target genes from a GWAS locus is challenging because most of the risk-associated variants are in non-protein-coding regions and do not alter amino acid sequences[10,11]. Instead, most GWAS variants appear to function as cis-regulatory elements (CREs) and alter gene expression levels by affecting the binding of trans-acting factors (e.g., transcription factor, TF) in gene promoters and enhancers[12]. Consistent with this hypothesis, approaches linking genetic variants to their putative targets of transcriptional regulation have been powerful in identifying susceptibility genes from GWAS loci. For example, expression quantitative trait loci (eQTL) approach can detect the association between GWAS variants and transcript levels of nearby genes. Chromatin interaction approach can link GWAS variants located in candidate CREs (cCREs) to target gene promoters, using physical interaction via chromatin looping[13]. However, the regulatory activities of the CREs are highly specific to cell types and cellular contexts[14], and the current bulk tissue-based eQTL and cell-line-based chromatin interactions data might not comprehensively capture these diverse biological contexts as well as the effect of environmental exposures on gene expression regulation.

Lung tissue has diverse cell types, and lung tumorigenesis is considered an interplay between the cells of cancer origin and their microenvironment, including immune cells, in the presence of external exposures such as smoking. Cells of lung cancer origin have been mainly traced in animal models, which suggested the roles of alveolar type II (AT2) and club cells for lung adenocarcinoma, basal cells for lung squamous cell carcinoma, and neuroendocrine cells for small cell lung cancer, which are all in the epithelial group of lung cells. Recent single-cell approaches using human and mouse lung tissues identified transient states of AT2 cells considered as stem/progenitor cells[15–19], hinting at their potential involvement in lung adenocarcinoma. However, many of these epithelial cell types of presumed cancer origin are rare in lung tissue (<5%) based on multiple classical studies[15]. Furthermore, in the fresh-dissociated single-cell suspension of lung tissue, epithelial cells tend to be depleted while immune cells are overrepresented[15]. Given this complexity, even when bulk-tissue-based eQTL identified multiple candidate target genes, it is difficult to determine in which cell types (e.g., cells of cancer origin vs. immune cells) the lung cancer susceptibility genes might be functional. Moreover, complex interplay of multiple target genes potentially regulated by one or more risk-associated variants in different contexts has not been explored.

To this end, single-cell multiome approaches combining single-nucleus RNA-sequencing (snRNA-seq) and single-nucleus assay for transposase-accessible chromatin with sequencing (snATAC-seq) can provide a powerful alternative to annotating GWAS loci[20,21]. While there are abundant single-cell based resources for human lung tissues as recently compiled by Human Lung Cell Atlas (HLCA)[18], single-cell profiling of chromatin accessibility in human lung tissues is still limited to a small number of datasets[22] and to our knowledge a joint profiling of transcriptome and chromatin accessibility in the same barcoded cells has not been performed. Moreover, cell-type-specific gene expression changes by lung cancer-relevant exposures such as smoking have not been formally profiled in human lung tissues.

In this work, we perform a barcode shared snRNA-seq and snATAC-seq of human lung tissues while incorporating age/sex-matched Korean ever- and never-smokers and adopting a strategy to enrich epithelial cell population. This multiome approach enables joint clustering between the two modalities, identification of cell-type specific cCREs, and linkage of cCREs to target genes[23]. By capturing the lung cancer-relevant cellular contexts, we extend the functional characterization of lung cancer GWAS loci representing diverse populations.

## Results

### Single-cell multiome design representing lung-cancer-relevant contexts

To generate a single-cell dataset representing lung-cancer-relevant cellular contexts, we designed a study focusing on smoking status as the main environmental exposure and cells of lung cancer origin to capture the endogenous lineage-specific gene regulation. We collected tumor-distant normal lung tissues from sex- and age-matched ever- and never-smokers ($n = 8$ samples each), which were freshly dissociated and cryopreserved before further processing (Fig. 1A; "Methods"). All the baseline characteristics of these samples are presented in Supplementary Data 1. Our design adopted barcode-shared snATAC-seq and snRNA-seq to jointly profile chromatin accessibility and gene expression from each cell type of lung tissues (Fig. 1B). This approach allows variant-cCRE colocalization, cCRE-gene linkage, and characterization of lung cancer GWAS loci at the levels of variant, gene, and cell type (Fig. 1C, D). Given that immune cells tend to be over-represented in normal lung tissues[15] and epithelial cells are more vulnerable to dissociation process as well as freezing and thawing, we employed balancing of the lung cell groups using cell surface marker labeling followed by fluorescence-activated cell sorting (FACS) ("Methods"). Antibodies against EpCAM, CD31, and CD45 were used to sort the live single-cells into four major categories: "epithelial" (EpCAM$^+$CD45$^-$CD31$^-$), "immune" (EpCAM$^-$CD45$^+$CD31$^-$), "endothelial" (EpCAM$^-$CD45$^-$CD31$^+$), and "stromal" (EpCAM$^-$CD45$^-$CD31$^-$) (Fig. 1E). To enrich epithelial cells, which include cell types considered relevant for lung cancer origin, we collected maximum number of "epithelial" cells from each sample while keeping a substantial ratio of "immune", "endothelial", and "stromal" groups (Fig. 1E; "Methods"). Based on flow cytometry, our strategy resulted in 20.3–52.4% (median 38.8%) of estimated "epithelial" cells across 16 samples after enrichment, which is a significant improvement (~4-fold increase) compared to 4.3–23.6% (median 10.0%) before balancing (Fig. 1F). "Immune" (median 31.4%), "endothelial" (median 25.3%), and "stromal" cells (median 4.2%) were also estimated at expected proportions after balancing.

### Simultaneous profiling of single-nucleus chromatin accessibility and gene expression identified classic and transient lung cell types

Using the balanced lung cells, we performed nuclei isolation followed by barcode-shared snATAC-seq and snRNA-seq to generate single-nucleus chromatin accessibility and gene expression matrices ("Methods"). The detailed summary metrics of the sequencing results are provided in Supplementary Data 2. After filtering low-quality nuclei, likely empty droplets, and doublets ("Methods"), we obtained 117,911 single nuclei from 16 samples with high-quality chromatin accessibility and gene expression profiles (Supplementary Data 3). After correcting the batch effects, these nuclei were clustered and assigned into 23 cell types (Fig. 2A, Supplementary Data 4) based on both modalities ("Methods"). They represent epithelial (AT2, alveolar type I or AT1, club, ciliated, goblet, basal, AT1/AT2, and AT2-proliferating), immune (natural killer or NK, T, macrophage, monocyte, B, dendritic, and NK/T), endothelial (lymphatic, artery, vein, and capillary), and stromal (fibroblast, smooth muscle, mesothelial, and myofibroblast) cells. A total of 47,453 epithelial cells (40.2%) were identified (Fig. 2B), which closely matched the estimation based on flow cytometry, supporting the validity of our enrichment strategy and cell-type annotation. We identified 36,308 immune (30.8%), 28,395 endothelial (24.1%), and 5755 stromal cells (4.9%) (Fig. 2B), and all 23 cell types were detected in 14 or more samples (Supplementary Data 5), supporting that the cell-type identity is not driven by a single or a few samples with batch effects. Consistent with published studies of human lung, we observed cell-type-specific canonical gene expression markers for all 23 cell types (Fig. 2C, additional markers in Fig. S1). Concurrently, we detected a total of 330,453 chromatin accessibility peaks (cCREs) across 23 cell types

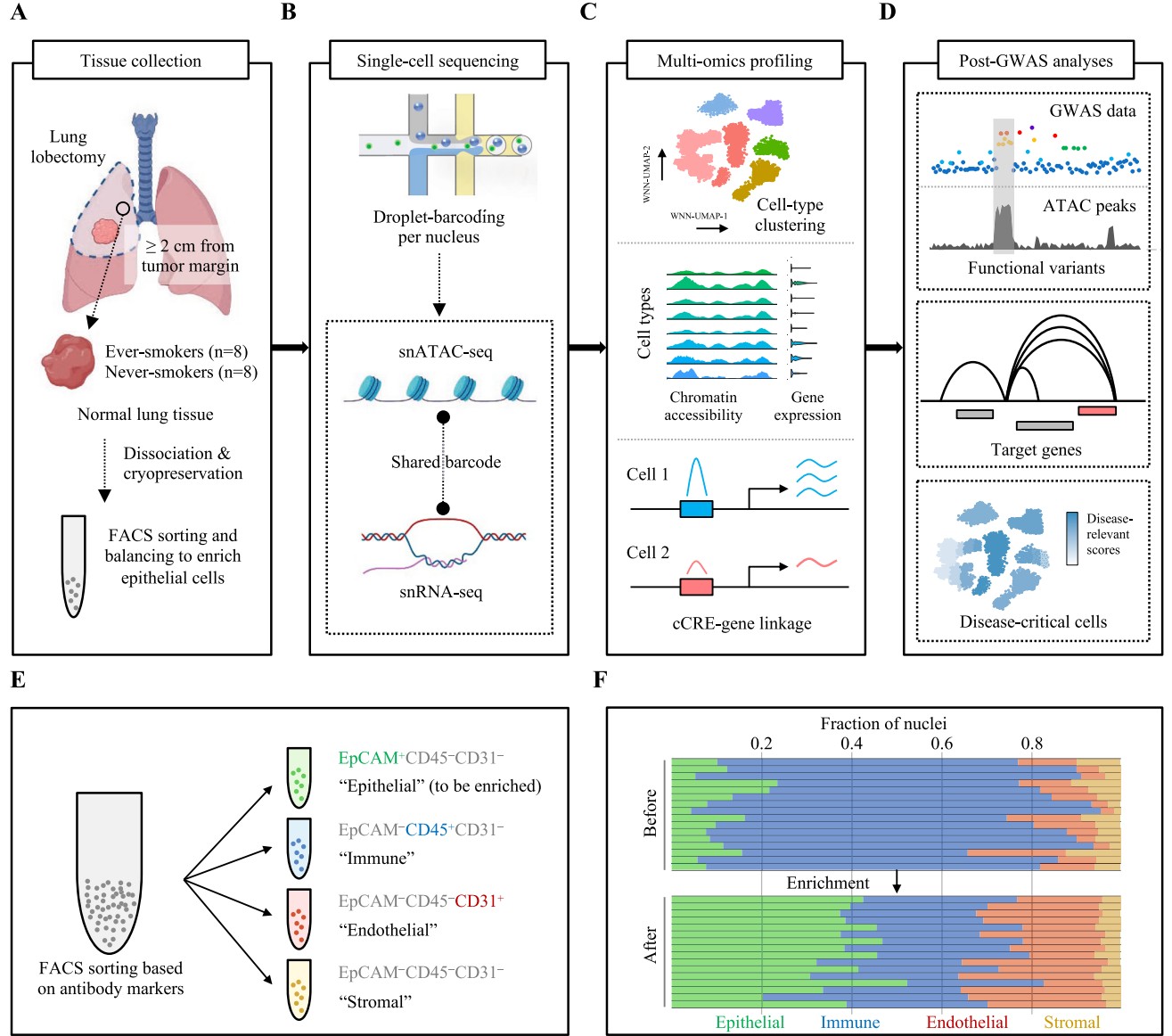

**Fig. 1 | Graphic summary of study design, workflow, and enrichment strategy.** Overview of the study pipeline, including tissue collection (**A**), single-cell sequencing (**B**), multiome profiling/analyses of chromatin accessibility/gene expression (**C**), and post-GWAS analyses of functional variants, target genes, and disease-relevant cells underlying lung cancer susceptibility loci (**D**). **E** Antibody markers of EpCAM, CD31, and CD45 were used to sort the live single-cells: EpCAM⁺CD45⁻CD31⁻ ("epithelial"), EpCAM⁻CD45⁺CD31⁻ ("immune"), EpCAM⁻CD45⁻CD31⁺ ("endothelial"), and EpCAM⁻CD45⁻CD31⁻ ("stromal").

Epithelial cells were enriched by balancing the ratios among "epithelial", "immune", "endothelial", and "stromal". **F** Bar plot shows the nuclei fraction of "epithelial" (green), "immune" (blue), "endothelial" (red), and "stromal" (yellow) before and after enrichment. Each bar represents an individual sample following the order of sample IDs listed in Supplementary Data 1. Source data are provided as a Source Data file. **A**, **B** were created with BioRender.com released under a Creative Commons Attribution-NonCommercial-NoDerivs 4.0 International license (https://creativecommons.org/licenses/by-nc-nd/4.0/deed.en).

using snATAC-seq data. To validate the consistency between chromatin accessibility and the corresponding gene expression, we calculated gene activity scores[24] based on the chromatin accessibility of the promoter and gene body region ("Methods"). Among 41 canonical cell-type marker genes, 37 showed a correlation (Pearson correlation coefficient $R > 0.5$, $P < 0.01$) between the percentages of cells expressing a marker gene and those with a non-zero gene activity across the cell types, and 17 of them showed a stronger correlation (Pearson correlation coefficient $R > 0.8$, $P < 1 \times 10^{-5}$; Fig. S2). Moreover, the co-embedding of the cells identified from snRNA-seq and snATAC-seq data showed substantial overlap between the two modalities (Fig. S3). We then surveyed the characteristics of the detected cCREs. Similar to the observations in previous snATAC-seq and multiome studies in different tissue types[20,25,26], a substantial proportion of the cCREs were detected only in

a single cell type ($n = 121,088$; 36.6%) (Fig. S4A) or in a single category ($n = 258,477$; 78%). Most of these cCREs overlapped with genic or gene promoter regions, including promoter (22%), exonic (4.3%), and intronic regions (46.1%) (Fig. S4B). The cCREs detected in all 23 cell types displayed a higher proportion of promoter-overlapped cCREs (91.8%) compared to those in the cell-type private ones (15.9%). Conversely, cell-type private cCREs displayed a higher proportion of intronic (48.4%) and intergenic cCREs (29.6%), which is consistent with the roles of distal enhancers in cell-type-specific gene regulation[25].

Notably, we identified a rare cell type, AT2-proliferating cells (0.13%), which expressed an AT2 marker (*SFTPD*) as well as the markers of cell proliferation, *STMN1*, *TYMS*, *TOP2A*, *CDK1*, and *MKI67* similar to two previous studies ("AT2-proliferating" and "cycling-AT2")[18,19] (Fig. 2D) and the HLCA annotation (Fig. S5). Consistent with the gene

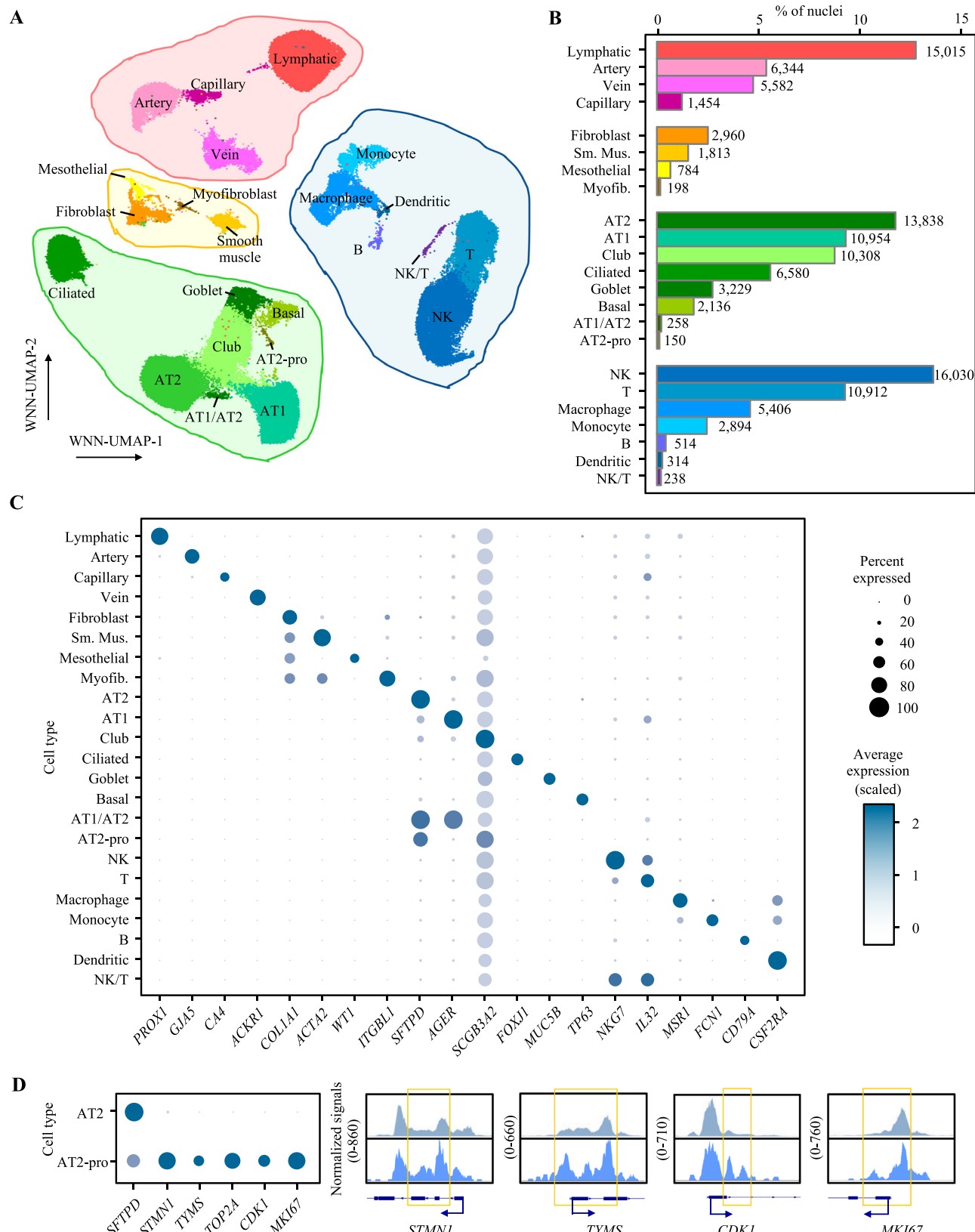

expression data, we observed distinct chromatin accessibility around the promotor-adjacent enhancer regions of *STMN1*, *TYMS*, *CDK1*, and *MKI67* in AT2-proliferating cells when compared to those in AT2 cells. To validate this finding, we utilized immunohistochemistry (IHC) and detected Ki-67 co-stained with SFTPD in a small subset of SFTPD-expressing cells in lung alveoli (Fig. S6).

**The intercellular communications of human lung between ever- and never-smokers**

Among 117,911 total nuclei, 59,250 were from ever-smokers and 58,661 were from never-smokers. We first compared the gene expression between ever- and never-smokers in each cell type to nominate the "smoking-responsive genes" using a stringent pseudobulk method

**Fig. 2 | Identification of major cell types of the human lung via joint profile of snRNA-seq and snATAC-seq. A** Weighted Nearest Neighbor (WNN) clustering results of the 117,911 single-nucleus using the profiles of chromatin accessibility and gene expression after quality controls with cell-type annotation based on canonical markers, which resulted in 23 cell types across endothelial (red shades), stromal (yellow shades), epithelial (green shades), and immune (blue shades) cells.
**B** Fraction of nuclei of each cell type relative to the total number of nuclei. Numbers next to each bar denote absolute counts out of 117,911 nuclei. **C** Dot plot visualizing the normalized RNA expression of selected marker genes by cell type. The color and size of each dot correspond to the scaled average expression level and fraction of expressing cells, respectively. Additional markers are shown in Fig. S1. **D** The dot blot on the left visualizes the normalized RNA expression of AT2 and AT2-

proliferating cells (AT2-pro). The color and size of each dot correspond to the scaled average expression level and fraction of expressing cells, respectively, with the same scale and fraction defined in (**C**). The sequencing tracks on the left visualize chromatin accessibility signals around selected marker genes by cell type. Each track represents the aggregate snATAC signal of all three included cell types normalized by the total number of reads in the transcription start site (TSS) region. Arrows show the transcriptional directions of the genes. Coordinates for each region are as follows: *STMN1* (chr1:25904993-25908229), *TYMS* (chr18:657083-658911), *CDK1* (chr10:60768638-60777926), *MKI67* (chr10:126261833-126268756). Yellow squares highlight the main differences of chromatin accessibility between two cell types. Source data are provided as a Source Data file.

("Methods"). While only one gene (*AC017002.5*, FDR = 0.0066, lymphatic cells) passed the multiple testing correction cutoff (FDR < 0.05), 24 genes displayed suggestive differences at a relaxed cutoff ($P < 1 \times 10^{-4}$), including previously implicated genes in smoking and related traits (Supplementary Data 6). Similarly, we compared the chromatin accessibility between ever- and never-smokers in each cell type to nominate "smoking-responsive cCREs" ("Methods"). A total of 1677 unique "smoking-responsive cCREs" were identified (FDR < 0.05, logistic regression, Supplementary Data 7; Fig. S7), including those within the promoter or intron of suggestive smoking-responsive genes with a matching direction (Supplementary Data 6), which corroborated the findings in gene expression.

We then asked whether cell-cell communication is different based on smoking status since smoking has been associated with cellular processes that involve intercellular interactions such as inflammation and immune functions[27]. Based on the known ligand-receptor pairs in CellChatDB, significant cell-cell communications were detected at the pair level and then summarized at the pathway level, for the comparison between the cells from ever- and never-smokers ("Methods"). We nominated the top five pathways displaying the largest differences in the communication strength based on smoking status across all cell types. The top five pathways with elevated communication strength in ever-smokers (MHC-I, UGRP1, CD46, NRXN, NEGR) and in never-smokers (MHC-II, VCAM, CD6, ALCAM, THBS) displayed the differences both at the level of individual cell types and when summarized across all cell types (Fig. S8). Notably, Major Histocompatibility Complex (MHC)-I and -II pathways exhibited an opposite trend across all cell types between ever- and never-smokers. This observation is, in principle, consistent with a recent study reporting that the levels of *HLA-A, B, C* (MHC-I) were higher in the lung adenocarcinoma cells from smokers compared to never-smokers, and MHC-II genes were elevated in cell subclusters representing never-smoker cancer cells[28]. MHC-II genes (*HLA-DRA*, *HLA-DRB1*) were also suppressed by smoking in multiple types of airway epithelial cells in a previous study[29]. To further validate the opposing trend of MHC-I and -II pathway communications in an independent dataset, we compiled normal human lung samples from 35 ever- and 27 never-smokers (248,380 cells across 21 matching cell types; HLCA[18]). We observed that MHC-I pathway communications were relatively higher in ever-smokers across all the cell types, while MHC-II pathway communications were higher in never-smokers mainly in macrophage, B, and dendritic cells (Fig. S9). Although exploratory, our findings provided a descriptive summary of cell-cell communications across the lung cell types between ever- and never-smokers.

### Epithelial and immune cell-specific *cis*-regulation underlies lung-cancer-associated variants

Using the cCREs that we detected in each cell type of the lung, we aimed to better understand the risk loci identified by GWASs of lung cancer. Combining the four most recent lung cancer GWASs representing European, East Asian, and African ancestries as well as major lung cancer histological types and smoking status[9,30–32], we compiled a set of candidate causal variants (CCVs) (Fig. 3A). We included 51 non-

overlapping GWAS loci (Supplementary Data 8) and 2574 unique CCVs (Supplementary Data 9) based on the GWAS statistics and linkage disequilibrium (LD) (log-likelihood ratio 1:1000 or $R^2 \geq 0.8$ to the lead variant in the study population). To characterize how these CCVs align with cCREs, we performed CCV-cCRE colocalization followed by allelic TF abundance and footprinting analyses (Fig. 3B).

First, we observed that 323 CCVs (12.5% of the tested CCVs) of 35 GWAS loci (68.6% of the GWAS loci) colocalized with a cCRE from one or more cell types (Supplementary Data 10), which indicated that a substantial proportion of GWAS loci was covered by this approach and a considerable variant prioritization was achieved. To nominate the lung cell types where lung cancer GWAS signals are enriched the most, we first integrated GWAS and snATAC-seq data using SCAVENGE[33]. SCAVENGE directly uses GWAS variants that colocalize with cCREs as opposed to indirectly linking the variants to gene expression data based on the distance to nearby genes or bulk-based expression correlation. We calculated the trait relevance scores (TRS) for each cell and compared them between cell types and cell categories ("Methods"). At the level of cell categories, epithelial and immune cell types showed higher mean TRS compared to endothelial and stromal cell types (3.60 and 3.57 vs 2.71 and 2.84, respectively) (Fig. S10). Notably, AT2 cells and AT2-proliferating cells displayed the highest mean TRS, which is consistent with a previous study pointing to AT2 cells for showing a suggestive enrichment of lung adenocarcinoma susceptibility based on lung single-cell RNA-seq (scRNA-seq) data[18]. We then defined the cell-type specificity for each CCV-colocalized cCRE by assessing whether the cCRE was being exclusively called in specific cell types or categories ("Methods"). We noted that 61% of the CCVs colocalized with cCREs were detected in a single cell-type category, with 28% to epithelial, 20% to immune, 9% to endothelial, and 4% to stromal (Fig. 3C). For example, in loci such as 30_9p13.3 and 19_6p21.1, we observed CCV-colocalizing cCREs that were mainly detected in epithelial category and in 6_2q33.1 and 5_2p23.3, those mainly observed in immune category (Fig. S11). These category-specific CCV-colocalized cCREs were significantly enriched in immune cells (adjusted $P = 0.0198$, Hypergeometric test; Fig. S12A), which was consistent with the observation in SCAVENGE analysis. Notably, there were substantial proportions of CCVs that were assigned to a single cell type (Fig. 3C). Among them, macrophage- and AT2 cell-specific cCREs colocalized with the largest numbers of CCVs (Fig. S12B). Cell-type-specific CCVs were also detected in rarer cell types (e.g., mesothelial and dendritic), which account for less than 1% of the total cell number. Although not cell type-specific, 17 CCV-colocalized cCREs from 12 loci were detected in AT2-proliferating cells, suggesting the contribution of transient epithelial cell types in lung cancer susceptibility (Supplementary Data 10). At locus level, we observed 5 loci that were assigned to epithelial category and 4 that were assigned to immune category (Fig. 3D). These results suggested that epithelial and immune cells are important cell types underlying the lung cancer GWAS loci.

Among the cell-type specific CCV-colocalizing cCREs were those from rare lung epithelial cell types. In the locus 15_5p15.33, two CCVs (rs7726159 and rs7725218) colocalized with a cCRE within *TERT* which

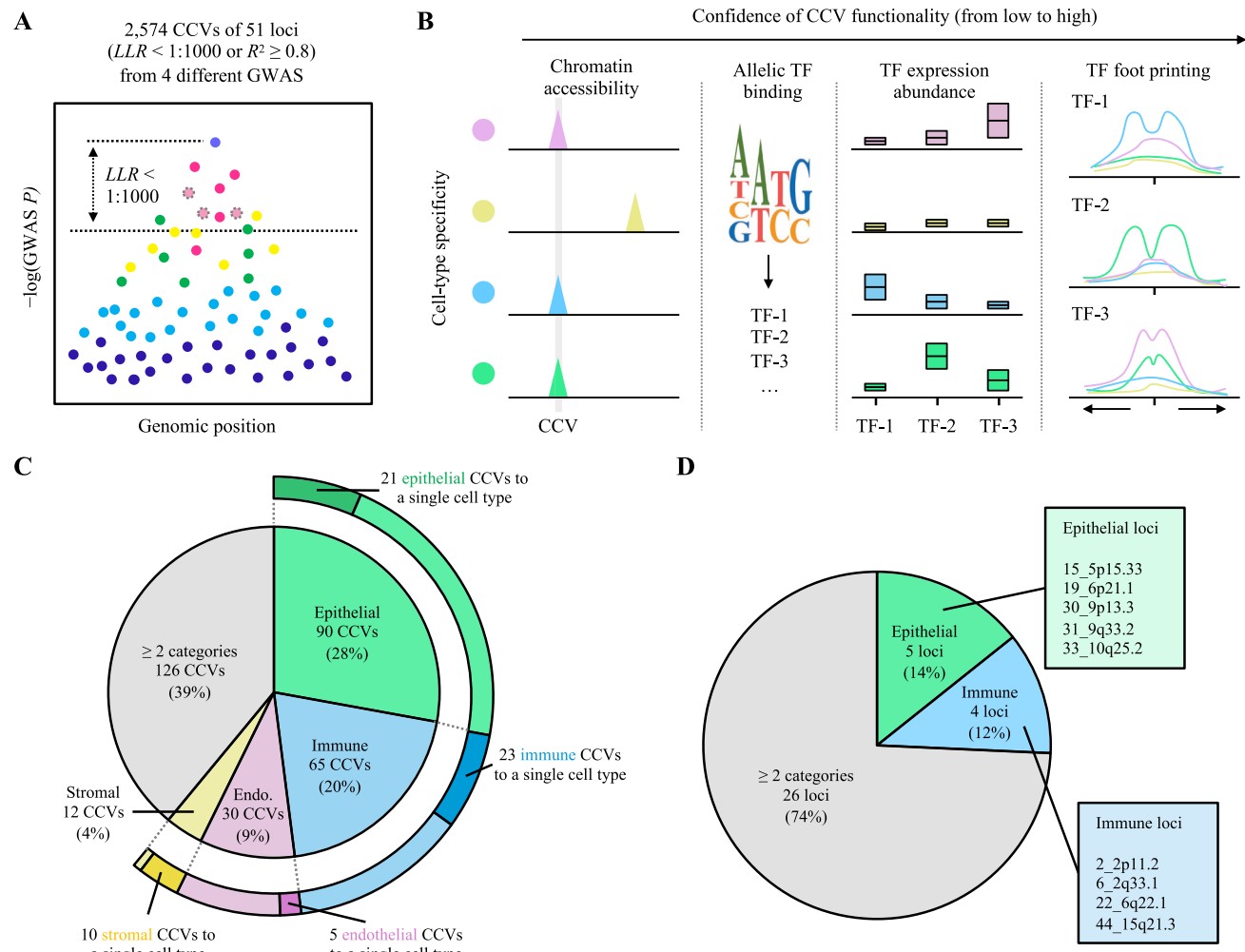

**Fig. 3 | Characterization of the cell-type specificity underlying lung-cancer-associated functional variants. A** Illustration of the candidate causal variants (CCVs) selection from GWAS loci. Each dot represents a variant, and the LD to lead variant is color-coded. Dots with dashed outline represent the variant in high-LD with the lead variant but not covered in the original GWAS summary statistics. LLR: log likelihood ratio (**B**) The confidence level of the CCV functionality is established by colocalizing them with accessible chromatin regions, predicting allelic TF binding effects, assessing TF abundance, and TF footprinting. **C** Piechart presents the fraction of CCVs assigned to different cell-type categories (inner pie) and assigned to a single cell type (external pie). **D** Pie chart presents the fraction of loci assigned to different categories.

is specific to basal cells, an epithelial cell type accounting for 1.8% of the total lung cells in our dataset (Fig. 4A). *TERT* expression is active in embryonic stem cells but silenced in differentiated cells; re-activation of *TERT* is central to cellular immortalization and tumorigenesis in multiple cancer types[34]. Consistent with this idea, basal cells in the epithelium of airway and lung are considered stem/progenitor cells with self-renewal ability[35]. IHC of our lung tissue samples displayed a weak co-staining of TERT in a subset of KRT17-positive basal cells, although TERT detection was not exclusive to basal cells (Fig. S13). To further examine the function of these two cCRE-overlapping CCVs, we performed reporter assays comparing the enhancer activity of lung cancer risk and protective alleles of each variant within a short sequence (145 bp) using A549 lung cancer cell line as a part of massively parallel reporter assays ("Methods"). The results demonstrated that one variant displayed significant allelic transcriptional activity at false discovery rate (FDR) < 1% with the lung cancer risk-associated allele showing higher levels, albeit with a modest effect size (rs7726159, $\log_2$fold change = 0.18, FDR = $1.76 \times 10^{-4}$), while the other was not significant (rs7725218, $\log_2$fold change = −0.05, FDR = 0.043) (Fig. 4B). These data suggested the potential utility of our dataset in detecting cell-type contexts of gene regulation underlying lung cancer susceptibility.

To further prioritize potential functional variants from the cCRE-colocalized variants, we performed TF analyses using cell-type-specific TF expression and TF footprinting. We first predicted TF binding affinity of each CCV (Fig. 3B), and 50% of the variants displayed allelic differences in binding scores to one or more TFs (Supplementary Data 10). For these allelic TFs, we investigated cell-type-specific expression levels. Fifty-six unique allelic TFs predicted for the CCVs across 20 loci were abundantly expressed in the same cell type(s) where the cCREs were detected (Fig. S14), suggesting a condition favorable for a binding event. We further performed TF footprinting for all the allelic TFs ("Methods"), and 82 unique TFs predicted for the CCVs across 27 loci exhibited an enriched average accessibility to their motif-flanking regions, suggesting a potential binding of these TFs to cCREs in the lung cells (Fig. S14). These data allowed us to score the functionality of the CCVs based on the cCRE overlap, allelic binding TF prediction, TF abundance, and footprints (Supplementary Data 10). Overall, 111 cCRE-overlapping CCVs from 29 GWAS loci were supported by either an average footprint detection or a cell-type matching TF abundance (functional score = 3), and 37 CCVs from 15 loci were supported by all four categories (functional score = 4, ranging from 1 to 7 CCVs per locus; median = 2), providing a substantial variant prioritization (Fig. S15;

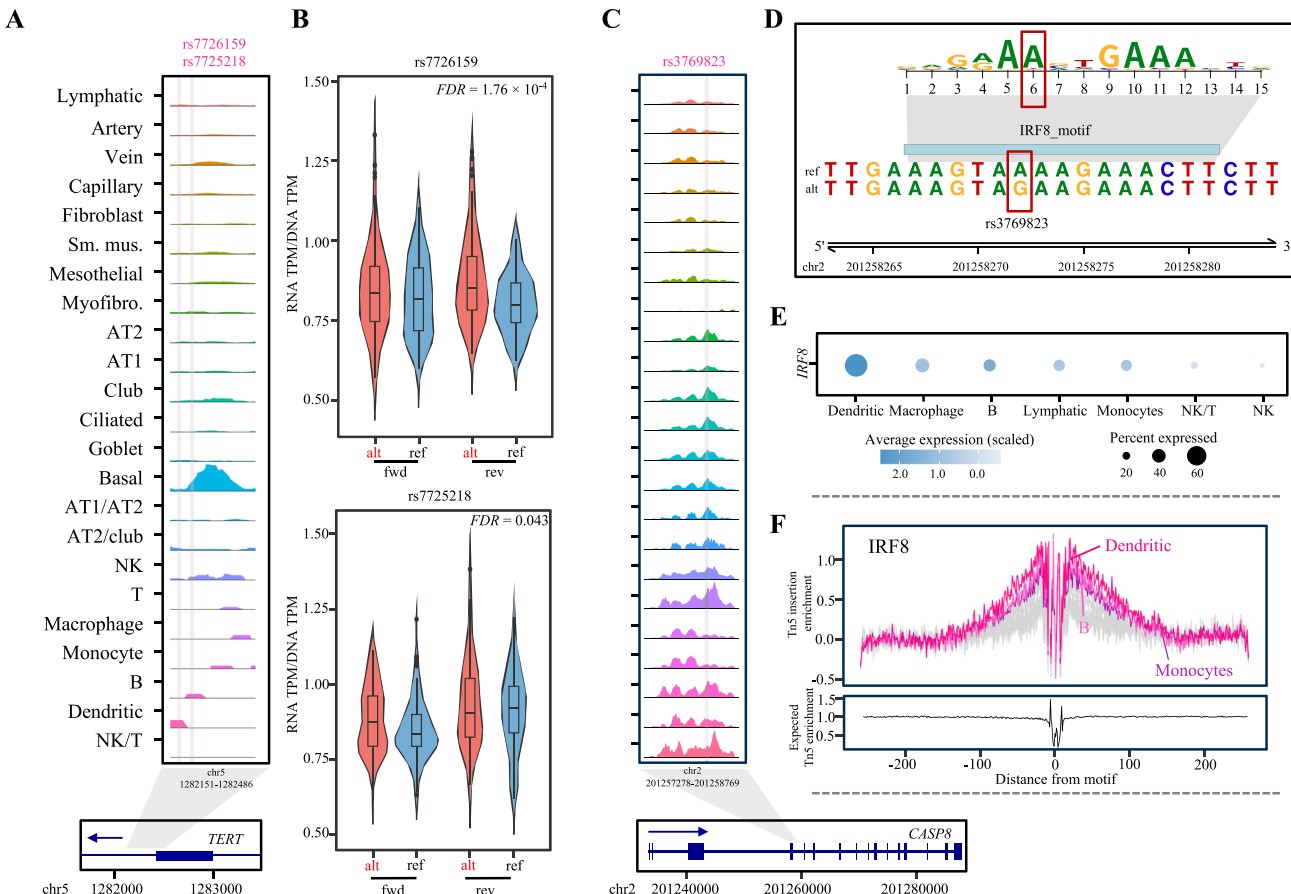

**Fig. 4 | Cell-type-specific variant function. A** Sequencing tracks representing chromatin accessibility near the CCVs marked by rsIDs and vertical pink lines. Each track represents the aggregated snATAC signal of each cell type, normalized by the total number of reads in the regions (y-axis scale 0–71). Arrow depicts the transcriptional direction of *TERT*. **B** Normalized transcriptional activity of 145 bp sequences encompassing rs7726159 (upper) and rs7725218 (lower) tested by massively parallel reporter assays in A549 lung cells. TPM: tags per million, alt: alternative allele (lung cancer risk-associated), ref: reference allele, fwd: forward, rev: reverse. Center lines show the medians; box limits indicate the 25th and 75th percentiles; whiskers extend 1.5 times the interquartile range from the 25th and 75th percentiles, outliers are represented by dots. Density is reflected in the width of the shape. Number of tags combined from five biological replicates for statistical testing are *n* = 125, 120, 120, and 110 for rs7726159 and *n* = 105, 110, 105, and 115 for rs7725218, from left to right. FDR values were calculated by the Wald test and corrected by the Benjamini-Hochberg procedure. **C** The sequencing tracks of chromatin accessibility, CCVs, and gene transcriptional direction using the same style as (**A**) (y-axis scale 0–490). **D** Position weight matrix of IRF8 motif shown as the height of the motif logos with the genomic location at the bottom. The variant position (rs3769823) within the motif is indicated with a red box. **E** The normalized mRNA expression of *IRF8* across 7 cell types with the highest *IRF8* expression. The color and size of each dot correspond to the scaled average expression level and fraction of cells expressing *IRF8*, respectively. **F** The upper part displays the average footprint profile of the IRF8 across all detected peaks in each cell type. Three cell types with the highest average footprint profiles for IRF8 motif are shown in shades of pink and the remaining cell types in gray. Tn5 insertion bias was corrected by subtracting the Tn5 signals from the average footprint signals. The lower part shows the expected Tn5 enrichment based on distance from motif. Source data are provided as a Source Data file.

Supplementary Data 10). Among them was a previously identified multi-cancer-associated functional variant, rs3769823, colocalized with a multi-cell-type cCRE within a *CASP8* alternative promoter (Fig. 4C). This missense variant was shown to alter CASP8 protein activity and affect apoptosis and proliferation of lung cancer cells[36] but also displayed a strong allelic transcriptional activity in immortalized melanocytes and melanoma cells[37]. Among many TFs predicted to display allelic binding affinity to this variant, *IRF8* was abundantly expressed in dendritic cells, a cell type where the average footprint profiles of its binding motif were also detected (Fig. 4D–F; Supplementary Data 10). Identification of a known functional variant provided support for our TF abundance and footprinting approach for variant prioritization as well as further insights into potential cell-type specific roles of a known susceptibility gene. Our dataset identified lung cancer-relevant cell types of the lung and nominated lung cancer CCVs that might be under cell-type-specific regulation, including a subset involving potential allelic TF binding.

## Multi-level linkage identified context-specific candidate susceptibility genes from lung cancer loci

We leveraged the barcode-matched multiome data to identify target genes of the lung cancer CCV-colocalized cCREs by employing a multi-level cCRE-gene linkage ("Methods"). Specifically, we first performed a "cCRE module" analysis to find a group of cCREs displaying co-accessibility and assign them a unique cCRE module membership. We then performed cCRE-cCRE and cCRE-gene correlation analyses to identify the cCREs or cCRE modules whose accessibility is significantly correlated with the accessibility of a promoter cCRE and/or the expression of a gene in *cis* (±1 Mb) ("Methods", Fig. 5A). We reasoned that a direct correlation between a cCRE accessibility and a gene expression provides the strongest evidence compared to a link through a highly correlated cCRE (co-accessible score > 0.5) or a cCRE module membership (co-accessible score > 0.32). We also considered the location of the cCREs within a gene promoter (distance to a transcription start site, TSS) or when a cCRE or a cCRE module is correlated with a promoter cCRE without direct evidence of expression level

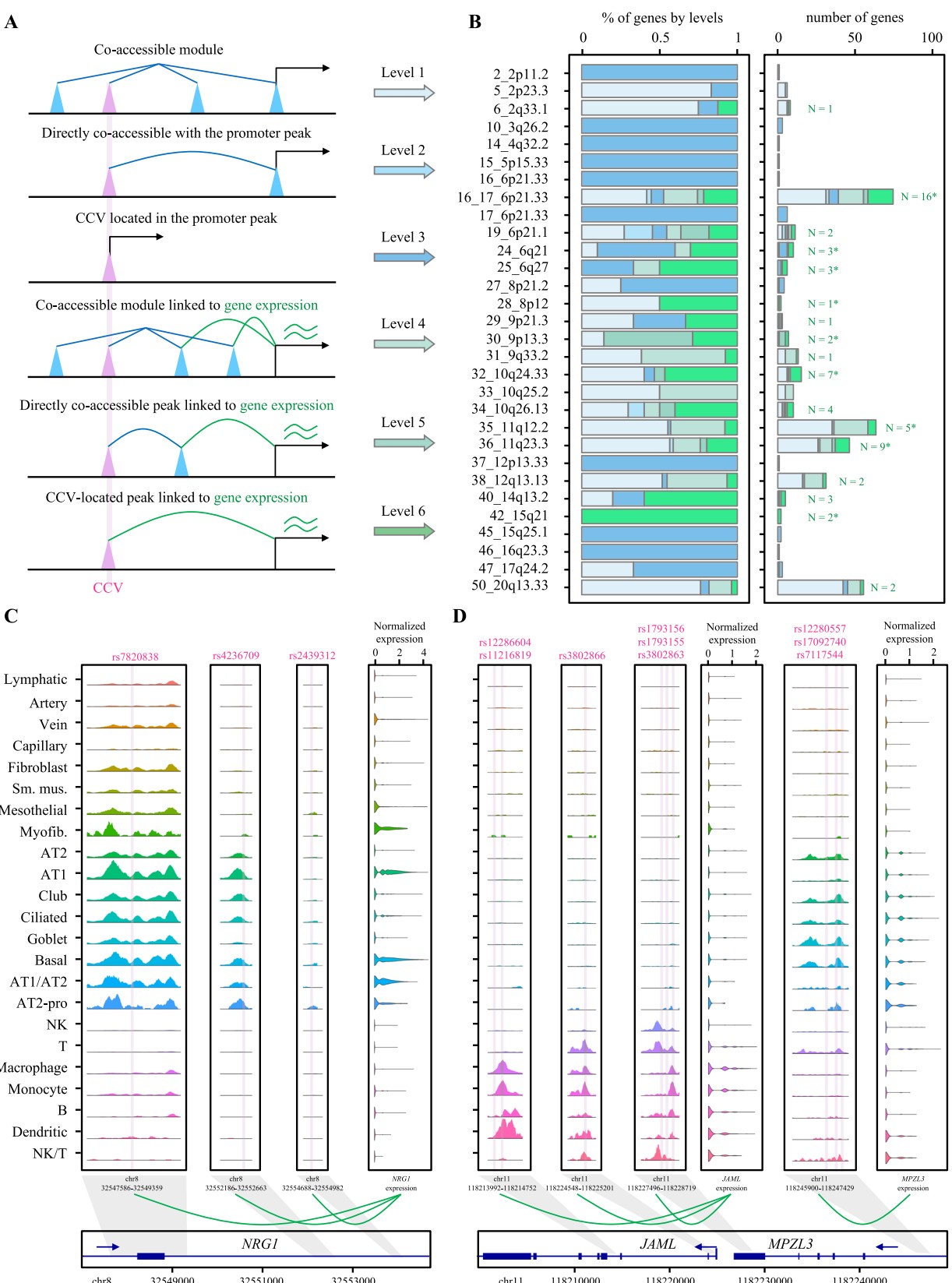

correlation albeit at a lower priority level. This multi-level system provided six different tiers of cCRE-gene linkage, with level 6 the strongest evidence. A total of 401 genes from 29 GWAS loci were linked to lung cancer CCV-colocalizing cCREs at levels 1 through 6 (Supplementary Data 11). Among them were 64 linked genes from 18 loci with the strongest evidence at level 6 (Fig. S16; Supplementary Data 12). Ten

of these 18 loci had one or more genes identified through bulk-tissue eQTL-based colocalization or transcriptome-wide association studies (TWAS) from published studies[9,32,38,39] (Fig. S17), while 17 loci had one or more genes that have not been identified before (Figs. 5B; S16). Notably, a CCV-colocalizing cCRE was linked to one to eight genes at level 6 (median 1 gene), and 82% of these cCREs were also linked to the

**Fig. 5 | Linkage between cCREs and genes and representative loci showing context-specific genetic regulation mechanisms. A** Illustration of the rationale in linking the cCREs to genes in different levels. cCRE module is represented by straight blue line. CCV-colocalizing cCREs are shown as pink cones. Highly co-accessible cCREs are represented by blue loop. The cCRE-gene correlations are represented by green loop. **B** The left panel presents the proportion of genes in different levels across GWAS loci. The right panel presents the number of genes in different levels across GWAS loci. The numbers refer to the level-6 gene numbers.

Asterisk indicates genes identified by TWAS or colocalization from previous studies. **C, D** The rsIDs of the CCVs are presented in the upper part of the sequencing tracks of chromatin accessibility with their locations marked in the tracks using vertical lines. Each track represents the aggregate snATAC signal of each cell type normalized by the total number of reads in the TSS regions (normalized values for the left three ATAC tracks: 0–380; normalized values for the right four ATAC tracks: 0–240). Arrows show the transcriptional directions of the genes. Source data are provided as a Source Data file.

same gene at level 4 or 5, indicating a redundant connection to target genes via modules of cCREs. Only 13% of the CCV-colocalizing cCREs were also located in the promoter of the same gene, suggesting that most of the CCV-colocalizing cCREs might regulate the target genes as distal enhancers. 74% of these putative distal enhancers were only detected in a single category of cell types. While +/-1Mb *cis*-window likely covers main gene regulatory interactions given a median size of topologically associating domain being less than 500kb[40,41], we further explored a potential cCRE-gene linkage in wider ranges (±2 and 5 Mb), which nominated additional candidate susceptibility genes (Supplementary Data 13–16). These data indicated that our cCRE-gene linkage approach could be validated by tissue-based eQTL methods and further identify additional candidate lung cancer susceptibility genes.

For example, an epithelial cell-specific CCV-colocalizing cCRE at 30_9p13.3 locus was linked to *AQP3* based on levels 4, 5, and 6 (Figs. S11, S16; Supplementary Data 11). This locus was initially identified through a lung cancer TWAS[30,38] using lung tissue eQTL data (Fig. S17) and later became genome-wide significant in a larger GWAS[31]. Over-expression of *AQP3* in cultured lung fibroblasts leads to increased endogenous DNA damage levels[38]. Our data validated *AQP3* as a lung cancer susceptibility gene in this locus and suggested potential involvement of epithelial-specific cCREs in its regulation. In the 6_2q33.1 locus, two CCV-colocalizing cCREs are located in *CASP8*. While a multi-cell-type cCRE (Chr2: 201,257,278-201,258,769) was linked to *CASP8* based on level 1, 2, or 3 (Supplementary Data 11), an immune-cell-specific cCRE was linked to *CLK1* expression by level 4, 5, or 6 (Supplementary Data 10; Fig. S16). *CLK1* is ~400 kb away from the linked cCRE and encodes CDC2-like kinase 1 which is a splice factor kinase that has roles in tumorigenesis of multiple cancer types[42–44]. This locus is a multi-cancer-associated locus, and classically the role of *CASP8* in cancer cells has been mainly considered in their downregulation and consequent escape of apoptosis rather than immune-cell-specific roles[45]. Our data suggested the presence of potential cell-type specific target genes of this locus.

### Coinherited CCVs are linked to multiple cell-type-specific target genes

In addition to validating known genes and identifying previously unknown candidate target genes from lung cancer GWAS loci, our dataset allowed us to deconvolute complex cell-type- and context-specific gene regulation patterns by multiple coinherited lung cancer-associated variants.

First, most loci displayed a complex picture of potential context-specific gene regulation and multiple candidate susceptibility genes. Specifically, 9 of 18 loci displayed connections of level-6 candidate susceptibility genes with CCV-overlapping cCREs that were detected in two or more cell-type categories, while the other half displayed only category-specific gene connections (Fig. S16). All but one locus involved at least one category-specific cCRE connected to a target gene.

Second, we observed that multiple high-LD CCVs might individually affect the target gene expression via separate but correlated cCREs. In the locus 28_8p12, three CCVs colocalized with three different cCREs that are mainly observed in epithelial category. All three cCREs belong to the same cCRE module, and each of them individually

was linked to *NRG1* expression at level 6 (Fig. 5C). Consistent with this finding, these three cCREs overlapped with the region that was assigned to *NRG*1 based on the activity-by-contact chromatin interaction model[13] in lung cell lines. *NRG1* was also a TWAS gene in this locus using GTEx lung tissues eQTL dataset. Notably, all three CCVs are in high-LD with the GWAS lead variant ($R^2 = 0.99$–1, EUR), suggesting that multiple functional variants in LD might share the same target gene.

Third, we observed that distinct subsets of CCV-colocalizing cCREs could be linked to cell-type-specific target genes. For example, the 36_11q23.3 locus includes two clusters of CCVs, among others, that are linked to two separate level-6 genes in a cell-type specific manner (Fig. 5D). Namely, three cCREs colocalizing with six CCVs were mainly detected in immune-category cells and linked to *JAML* expression. Another cCRE colocalizing with three CCVs was mainly detected in epithelial-category cells and significantly correlated with *MPZL3* expression. A subset of CCVs in these two clusters was also supported by footprints of allelic-binding TFs (rs3802866, rs1793155, and rs12280557 with a variant score = 3; Supplementary Data 10). Notably, both *JAML* and *MPZL3* have been identified by eQTL colocalization and TWAS approaches using lung tissues[9,38], but it has not been clear which genes are expressed and thus functional in what lung cell types or contexts. These data suggested that both epithelial and immune-cell-specific lung cancer susceptibility genes could be functioning together based on the cellular contexts within the same lung cancer-associated locus.

## Discussion

In this study, we established a customized single-cell multiome dataset representing lung cancer-relevant environmental exposure and presumed cell types of origin. Using this dataset, we identified candidate functional variants and susceptibility genes for most published lung cancer-associated loci, many of which displayed cell type and context specificity. We also demonstrated the feasibility of using cryopreserved dissociated cells from solid tissues for single-cell applications by employing FACS sorting and cell-type balancing. Fresh tissue processing of surgically resected solid tissues is still a logistical challenge for many single-cell applications. While new technologies are emerging to use fresh-frozen or formalin-fixed paraffin-embedded tissues, our approach is an alternative that provides flexibility of cryopreserving dissociated tissues and enriching cell groups of interest by cell surface marker-based sorting.

Using single-cell multiome approach, we found that cCREs of lung tissue are mostly cell-type specific, and epithelial and immune cell groups contribute the most to genetic underpinning of lung cancer risk. Single-cell-based TRS assessment highlighted AT2 cells and AT2-proliferating cells, among epithelial cell group, as the top contributing cell types to lung cancer susceptibility. This finding is consistent with the previous understanding of AT2 cells being an origin of lung adenocarcinoma and also aligns with the hypothesis that rare stem/progenitor populations responsible for alveolar regeneration might be important in lung tumorigenesis. In addition, we provide evidence for the long-regarded importance of immune cell populations in lung tumorigenesis. While a recent LD score regression analysis based on lung single-cell expression data suggested the importance of AT2 cells in lung adenocarcinoma heritability[18], previous LD score regression

results using bulk tissues and cell-line-based annotation datasets hinted at the roles of immune cells in lung cancer susceptibility, especially in European smokers[5,32]. Immune cell populations could interact with tumor cells originating from epithelial cells to promote their growth and selection through inflammatory signaling, for example, but they could also suppress tumor cell growth via immune surveillance during the trajectory of tumorigenesis. The potential interplay of epithelial and immune cell populations could be observed in multiple lung cancer loci; CCVs from 14 loci overlapped with both epithelial-specific and immune-specific cCREs. Multiple examples from cCRE-gene linkage also suggested that different subsets of CCVs might regulate epithelial or immune-cell-specific target genes in a single locus.

CCV-cCRE colocalization enabled annotation of lung cancer GWAS loci based on rare or hard-to-culture cell types from normal lung tissues. For example, AT2 and proliferating sub-populations of AT2 cells are difficult to culture, making it challenging to investigate variant functionality for the genes that are specifically expressed in these cell types. Furthermore, we observed that long-suspected roles of candidate genes aligned with their cellular contexts based on the cCREs specific to rare lung cell types. The 5p15.33 locus near *TERT* is one of the strongest lung cancer signals and associated with multiple cancers. While previous studies in the context of multiple cancer types identified functional variants regulating *TERT* expression, due to its inactivation in normal differentiated tissues, eQTL of *TERT* has not been detected in the GTEx lung tissues. By inspecting a rare stem-like cell population of lung epithelium, basal cells (1.8%), we show that a subset of CCVs in this locus overlaps with a cCRE distinct to this cell type. Due to the small number of cells in this population of cells, cCRE-gene linkage was not observed for *TERT* expression, although a weak TERT co-staining was detectable in a subset of basal cells by IHC of lung tissues. Notably, the risk allele of one of the colocalizing variants displayed higher transcriptional activity in A549 lung cancer cells, although the effect size was modest. It is conceivable that multiple other context-dependent CREs could influence *TERT* expression in lung tissues.

Our dataset provided additional insights to potential mechanisms of multiple high-LD CCVs regulating their target genes in a context-dependent manner. While conventional approaches using bulk tissue or cell lines such as eQTL colocalization/TWAS and chromatin interaction analyses typically identify multiple target genes from a single GWAS locus, it is difficult to prioritize and interpret what genes are functional in what contexts. In our prime examples, we observed that some of these cases could be explained by cell-type-specific gene regulation via different subsets of trait-associated variants. In the 36_11q23.3 locus, *JAML* expression was correlated with three different CCV-colocalizing cCREs in immune cell types while the cCRE are almost undetectable in epithelial cell types. Conversely, epithelial cell-specific CCV-colocalizing cCRE was linked to epithelial cell-preferential expressed genes including *MPZL3*, which is adjacent to *JAML*. *JAML* encodes a junctional adhesion molecule playing a role in defense against infection, tissue homeostasis and repair, and inflammation in the tissue-resident gamma-delta T cells at the epithelial barrier[46,47], suggesting its potential roles in lung tumorigenesis through interplay with epithelial cells. We also show in the example of *NRG1* that multiple high-LD variants that might be indistinguishable in statistical approaches can be strongly associated with the same target gene while located in three separate cCREs. *NRG1* encodes Neuregulin-1, a ligand containing epidermal growth factor (EGF)-like receptor tyrosine kinase (RTK)-binding domain, and somatic fusion of *NRG1* is a rare tumor driver event but most frequently observed in lung adenocarcinoma cases among all cancer types[48].

Our descriptive cell-cell communication analysis suggested the MHC-I and MHC-II pathways displayed an opposite direction of changes between ever- and never-smokers. Anecdotal findings based on average gene expression levels of certain lung adenocarcinoma cell clusters were observed to a similar effect in a previous study[28], and MHC-II gene suppression was also shown in airway epithelial cells of smokers without a reported lung cancer[29]. A recent study observed a correlation between HLA-II heterozygosity and reduced lung cancer risk in European smokers, suggesting an escape from immune surveillance as a mechanism of tumorigenesis[49]. Consistently, multiple lung cancer GWAS hits were identified in the MHC locus on chromosome 6, and distinct signals were observed based on different studies representing European populations of mainly smokers or East Asian populations including a substantial proportion of never-smokers[50]. Further studies will be needed to begin to understand the mechanistic connections between potential changes of MHC pathways in smokers and lung cancer development.

Our study has several limitations. First, the differentially expressed genes based on smoking status were mostly suggestive and did not pass the stringent cutoff. A larger study with well-documented exposure details could help improve the detection power. Second, our dataset and the main analyses do not represent all the detectable lung cell types that have been reported. It is possible that we missed specific epithelial cell types that are the most vulnerable (e.g., neuroendocrine cells). To begin to explore additional cell type detection, we performed subpopulation analysis for the 16 cell type clusters with ≥ 1000 cells ("Methods") and identified 55 potential subpopulations (Fig. S18). Among them were two T-cell subpopulations differentially expressing known markers of CD4 or CD8 T cells and two fibroblast subpopulations differentially expressing known markers of adventitial or alveolar fibroblasts (Fig. S19). These subpopulations can be used as resources for future studies investigating gene regulation in more diverse lung cell types. Third, our cCRE-gene linkage mainly considered the regulation within 1 Mb distance, which may potentially overlook the long-range regulation. To begin to explore this possibility we performed and shared the full results of cCRE-gene linkage using 2 Mb and 5 Mb ranges as a data resource (Supplementary Data 13–16).

In summary, our study provided a unique dataset to characterize cell-type and context-specific genetic regulation underlying lung cancer susceptibility.

## Methods

### Human subjects and tissue collection

Patient tissues were obtained under a protocol approved by Yonsei University Health System, Severance Hospital, Institutional Review Board (IRB 4-2019-0447, 4-2022-0706), and informed consent was obtained from each patient prior to surgery. All samples were collected primarily to study risk factors and prognoses of respiratory diseases. The donors consented to the secondary use of their samples for comprehensive research purposes, including providing data containing personally identifiable information such as sex, age, and the medical center. All experiments were performed following applicable regulations and guidelines. We collected tumor-distant normal lung parenchyma tissue samples from patients who underwent lobectomy with a curative aim for primary lung adenocarcinoma. Most of the patients presented stage IA adenocarcinoma, and tumor stages were comparable between the groups based on self-reported sex or smoking status; in each of the four groups, ≥75% of the patients presented stage I. Tumor-distant normal samples were obtained from the periphery of the lobe, more than 2 cm away from the tumor edge, which is unlikely to present tumor phenotype based on guidelines of the National Comprehensive Cancer Network. Dissociated tissues from four of the female never-smokers were combined to produce a single sample with similar total cell counts to other samples, which resulted in a total of 16 samples (8 each from ever- and never-smokers) (Supplementary Data 1). All the participating patients did not receive any other cancer treatment prior to surgery. Never-smokers were defined as individuals who have smoked <100 cigarettes in the lifetime. Ever-smokers were defined as individuals who have smoked ≥100 cigarettes

in the lifetime, and their smoking status was recorded (age to start/quit smoking, current/former smoker, and self-reported pack years). There were no significant differences of age between ever- and never-smokers, and the two groups included an equal number of male and female-derived samples based on self-reported sex. Detailed characteristics of the patients included in the study are in Supplementary Data 1.

## Statistics and reproducibility

The sample size of comparing 8 vs. 8 for assessing smoking effect was determined based on a recent scRNA-seq study that has successfully detected differential gene expression between smokers ($n = 6$) and non-smokers ($n = 6$) in human tracheal epithelium[29], although no statistical method was used. Smoking groups were blinded during the tissue dissociation step. Single-cell data was filtered to exclude low-quality cells as described in Quality control and filtering and Clustering and cell-type annotation section.

## Tissue dissociation and cryopreservation

Fresh tissue dissociation and cryopreservation conditions were optimized to maximize cell viability and facilitate proper preservation of diverse cell types. Specifically, digestion enzyme titration was performed and freezing media were compared based on epithelial cell retention measured by flow cytometry after dissociation and freezing/thawing, respectively. The tissues were collected at a ~ 2 cm³ size and put in MACS Tissue Storage Solution (Miltenyl Biotec cat.130-100-008) within 30 min of surgical dissection. Samples were kept at 4 °C and processed within 3 h of dissection. Tissues were chopped into 2 mm diameter pieces within a dissociation tube to reduce cell loss. Multi Tissue Dissociation Kit 1 (Miltenyl Biotec, cat. 130-110-201) and gentleMACS Octo Dissociator (Miltenyi Biotec, cat. 130-096-427) were used for dissociation of the tissue into single-cell suspension with a reduced amount of enzyme R (25% of the regular amount). Red blood cells were removed using Red Blood Cell Lysis Solution (Miltenyl Biotec cat.130-094-183). Single-cell suspension of samples was frozen using 90% FBS and 10% DMSO and stored in liquid nitrogen until further processing except for the time of transfer in dry ice.

## Single-cell processing and cell-type balancing

Given that immune cells were reported to be overrepresented in lung tissues[15] and epithelial cells are relatively vulnerable to freezing and thawing, we employed balancing of major cell groups of the lung using cell surface marker labeling followed by FACS sorting. Balancing of fresh-dissociated lung cells using FACS sorting has been reported by other groups[15,51], applying roughly 2:1 ratio of epithelial to immune cells and achieving up to >30% of epithelial cells. Specifically, we thawed and filtered the dissociated single-cell suspensions using a 70 μm Pre-Separation Filters (Miltenyi Biotec, cat. 130-095-823) to remove debris. The filtered cells were labeled with cell viability marker (1 μg/ml DAPI, ThermoFisher Scientific, cat. D21490) and 2.5 μl of antibody markers of EpCAM (ThermoFisher Scientific, cat. 25-9326-42), CD31 (BioLegend, cat. 303116), and CD45 (BioLegend, cat. 304006) in 100 μl total volume. Live single-cells (DAPI-negative) were sorted based on three gates: EpCAM⁺CD45⁻ (designated "epithelial"), EpCAM⁻CD45⁺ (designated "immune"), and EpCAM⁻CD45⁻ (designed "endothelial or stromal") on BD FACSAria Fusion Flow Cytometer (BD Biosciences) (Fig. S20). To enrich for epithelial cells, which are considered to have key roles in lung cancer etiology, we collected all "epithelial" cells from EpCAM⁺CD45⁻ gates and balanced the ratios to roughly 6:3:1 ("epithelial": "immune": "endothelial or stromal").

## Nuclei isolation and single-nuclei multiome sequencing

Nuclei isolation was performed based on "Low Cell Input Nuclei Isolation" protocol (CG000365-Rev C, 10X Genomics) with a modification of the 1X lysis buffer treatment time (3-second duration). The concentration and treatment time of the lysis buffer were optimized based on the quality of nuclei measured by a 4-level grading of nuclei from microscopic images (non-lysed, ruptured, fair, intact) to achieve the minimum numbers of non-lysed cells or ruptured nuclei and maximum numbers of intact and fair-quality nuclei. After processing all the samples following the optimized protocol, nuclei quality of all the samples was measured (~200 nuclei/sample), and no significant differences were observed between samples from ever- vs. never-smokers (median percentage of fair or intact nuclei 57.5% vs. 56.5%). Isolated nuclei were used for single-cell capture and sequencing library preparation using the Chromium Next GEM Single Cell Multiome ATAC + Gene Expression Reagent Kits following the manufacturer's guidelines (CG000338- Rev E, 10X Genomics, USA). The snATAC-seq and snRNA-seq libraries were sequenced on a NovaSeq6000 platform using the following cycles: snATAC-seq (Read 1 N, 50 cycles; i7 Index, 8 cycles; i5 Index, 24 cycles; Read 2 N, 49 cycles) and snRNA-seq (Read 1, 28 cycles; i7 Index, 10 cycles; i5 Index, 10 cycles; Read 2, 90 cycles). Each sample achieved >499 million paired reads for snATAC-seq and >372 million paired reads for snRNA-seq.

## Generating single-nucleus gene expression and chromatin accessibility matrices

Raw sequencing data were converted to fastq format using Cell Ranger ARC "mkfastq" function (10x Genomics, v.2.0.1). snRNA-seq and snATAC-seq reads were aligned to the GRCh38 (hg38) reference genome using STAR and quantified using Cell Ranger ARC "count" function (10x Genomics, v.2.0.1). The raw outputs from Cell Ranger ARC achieved 7878–14,005 snATAC median high-quality fragments per nucleus, 98.1% to 98.9% snATAC valid barcodes, 1905–2941 snRNA median genes per nucleus, and 92.2–95.1% snRNA valid barcodes across 16 samples (Supplementary Data 2). Gene expression and chromatin accessibility matrices were generated separately for each sample.

## Quality control and filtering

We used DropletQC (v.0.9) to remove "empty" droplets containing ambient RNA from the gene expression matrices[52]. Cell Ranger ARC-generated "possorted_genome_bam.bam" files served as input for DropletQC. The parameter "nuclear_fraction" was calculated using the function "nuclear_fraction_tags" with default parameters. Empty droplets were identified by visualizing the density of nuclear fraction and settling the cutoff according to the "peak" in low-nuclear-fraction droplets (Fig. S21). The resulting expression matrices were processed individually in R (v.4.1.3) using Seurat (v.4.0.6)[53] and Gencode v.27 for gene identification. From each sample, we excluded cells with less than 500 or more than 25000 nCount_RNA (number of RNA read counts), cells with less than 1000 and more than 70,000 nCount_ATAC (number of ATAC read counts), and cells with more than 10% of counts corresponding to mitochondrial genes. In addition, Scrublet was applied to identify and remove doublets with an expected doublet rate 10% based on the loading rate[54]. Jointly applying ATAC and RNA filters resulted in a total of 117,911 cells from 16 samples with high-quality measurements across both modalities. After quality control and filtering, all cells across samples were merged.

## Gene expression data processing

Filtered gene–barcode matrices were normalized with the "SCTransform" function of Seurat, and the top 2000 variable genes were identified. Gene expression matrices were scaled and centered using the "ScaleData" function.

## Peak calling, annotation, and gene activity score

The snATAC peak calling and annotation were performed following the Signac pipeline[55]. Specifically, peaks were called using MACS2 with default parameters after combining the reads of all the cells in each cell type to determine the genomic regions enriched for Tn5 accessibility

from snATAC fragments, resulting in 330,453 peaks in total. As previously reported, peak calling could be influenced by cell numbers, especially in rare cell types[56]. In our dataset, the total number of peaks within cell types displayed a weak but positive correlation with cell numbers (Pearson $R^2 = 0.36$, $P = 0.0026$; Fig. S22A). Notably, among the abundant cell types (proportion > 1%; 16 out of 23 cell types accounting for 97.9% of the total cell number) there is no significant correlation between cell numbers and total peak numbers (Pearson $R^2 = 0.08$, $P = 0.29$; Fig. S22A), indicating that the cell numbers might not affect peak calling efficiency for most of the cell types. Moreover, cell type-specific peak numbers displayed no significant correlation with the total peak numbers ($P = 2.38 \times 10^{-7}$, Wilcoxon signed-rank test, Pearson $R^2 = 0.011$, $P = 0.63$; Fig. S22B-C). Peaks were then annotated according to distance to protein-coding genes using ChIPseeker, with "tssRegion" setting of −3000 to 3000[57]. Term frequency inverse document frequency (TF-IDF) normalization on a matrix was performed using "RunTFIDF" function. The gene activity score was calculated by the chromatin accessibility of the promoter region (2000bp upstream of the TSS) and gene body (TSS to transcription end site) via the "GeneActivity" function of Signac v1.12.0.

## Clustering and cell-type annotation

Using the normalized gene expression data, we performed principal component analysis (PCA) with 50 PCs to compute and store. A uniform manifold approximation and projection (UMAP)-based approach was applied for expression matrices with the first 50 PCs and for chromatin accessibility matrices with the 2nd through 50th PCs (the first PC was excluded as this is typically correlated with sequencing depth). Both expression and chromatin accessibility matrices were corrected for batch effect using Harmony (v.1.2.0)[58]. A Weighted Nearest Neighbor (WNN) method[53] was applied to integrate the weighted combination of RNA and ATAC-seq modalities. The "FindClusters" function was applied for clustering using smart local moving (SLM) algorithm for modularity optimization at a resolution of 0.5. Clusters were annotated based on canonical marker genes reported by three independent studies[15,18,22]. Clusters expressing canonical marker genes from two or more different cell types were designated as putative doublets and excluded, after which re-clustering was performed using the same parameters. Clusters with no detected marker genes were also excluded, after which the dataset was also re-clustered. Subpopulations were identified for the 16 cell types with ≥ 1000 cells using the 'FindSubCluster' function. Three parameters ($\lambda$, $\theta$, $\sigma$) were used to optimize the batch effect correction power. Detailed parameters and resulting cell numbers of subclusters are presented in Supplementary Data 17. All the final subclusters were manually checked to ensure that they were not derived from a single batch. Subcluster information is shared through a Seurat object ("Data Availability"). For efficiency of cCRE detection, we performed the downstream analyses based on the 23 main annotated clusters but provided the subcluster information as a data resource.

## Co-embedding analysis between snRNA and snATAC modalities

The co-embedding analysis involved three main steps. First, a set of anchors was identified between RNA (reference) and gene activity score (query) via the "FindTransferAnchors" function of Seurat v.4.0.6. Second, based on the identified anchors we imputed RNA expression (i.e., gene activity score) from snATAC-seq data via the "TransferData" function, and then merged the two datasets. Third, PCA and UMAP were performed on the merged dataset to visualize the co-embedding dimensions. Top 2000 variable genes were used in the co-embedding analysis.

## Immunohistochemical staining

Fresh tumor-distant normal lung tissues were fixed in formalin, embedded in paraffin, and sectioned at a thickness of 4 μm.

Immunohistochemical staining was performed using the specific antibodies and conditions detailed in Supplementary Data 18. Digital images of the stained slides were acquired using a slide scanner (ZEISS Axioscan 7, ZEISS). Entire sections were examined, with a minimum of four images captured and evaluated from each section to assess the expression levels. Co-staining of two antibodies was evaluated in two adjacent sections of serially sectioned tissue. The expression levels were categorized based on intensity (strong: clearly visible with use of an ×2.5 or ×5 microscope objective lens, moderate: clearly seen in a ×20 objective lens, or weak: can be seen only with a ×40 objective lens) following a published guideline for diagnostic IHC[59].

## Analysis of smoking-responsive genes

Smoking-responsive genes for each cell type were inferred by pseudobulk differential gene expression analysis using DESeq2 (v1.41.1). We initially performed a pilot analysis using a method without adjusting within-sample correlation (e.g., Wilcoxon rank-sum test) and observed $p$-value inflation in highly-expressed genes which was confirmed by a label-swapping and permutation analysis. To avoid false positive findings in highly expressed genes and also be able to incorporate other covariates in the model[56] we chose a pseudobulk-based method. DESeq2 model assumes a negative binomial distribution[60], and the gene expression counts were aggregated for each sample before differential expression analysis to account for the within-sample cell-cell correlation. Sex was incorporated as the covariate in the model. Given that the sample size is limited (8 ever-smokers versus 8 never-smokers) and the method is highly stringent, we report suggestive smoking-responsive genes using a relaxed cutoff of $P < 0.0001$ before multiple-testing correction.

## Analysis of smoking-responsive cCREs

To nominate smoking-responsive cCREs, peaks were recalled within each group (i.e., never- and ever-smokers) of each cell type. For example, the AT2 cluster was divided into AT2-ever-smoker subcluster and AT2-never-smoker subcluster for peak calling using the same procedure described above. A total of 314,686 peaks were detected and tested for differential accessibility between ever- and never-smoker groups within each cell type. Logistic regression model adjusting for the total number of peaks of each cell was used employing "FindMarkers" in Seurat (test.use = "LR", latent.vars = "nCount_peaks"). $P$ values were calculated by likelihood ratio test, and Bonferroni correction was used for multiple testing corrections. Differentially accessible peaks passing 5% FDR were nominated as smoking-responsive cCREs. After dropping 5 peaks that were not aligning with cell-type-level peak calling, 1677 unique cCREs were identified across the cell types.

## Intercellular communication analysis

CellChat (version 1.6.0) was used to infer ligand-receptor interactions based on scRNA-seq data[61]. CellChat applies a signaling molecule interaction database (CellChatDB.human) to predict intercellular communication patterns based on differentially overexpressed receptors and ligands, including soluble agonists/antagonists as well as membrane-bound receptors/co-receptors. Specifically, a significant cell-cell communication was detected at the level of ligand-receptor pairs, based on a statistical test that randomly permutes the group labels of cells (permutation $P < 0.05$) and then recalculates the communication probability. Known ligand-receptor pairs from the CellChatDB database were used to compute the communication probability at the pair level, which was summed into communication strength at the pathway level. The differences of summed probabilities between ever- and never-smokers were used to nominate the top-ranking differential pathways. The KEGG pathway database (https://www.genome.jp/kegg/pathway.html) was used to infer the relationship between ligand-receptor pairs and pathways. The intercellular

communication analyses were performed separately using cells from ever-smokers and never-smokers for comparison at different levels. We nominated the top 5 pathways showing the largest difference in the relative sum of communication strength of all cell types (scale of 0–1) between ever- and never-smoker groups in each direction (top 5 increased and top 5 decreased in ever-smokers). An independent dataset[18] of 62 normal human lung samples was curated for validating the trend of MHC-I and MHC-II communication. The processed file for the integrated Human Lung Cell Atlas (HLCA) core dataset was downloaded from CZ CELLxGENE web portal (https://cellxgene.cziscience.com/collections/6f6d381a-7701-4781-935c-db10d30de293). From 107 individuals, 62 samples were selected based on the avail-ability of the smoking history information. Current and former smo-kers were combined and labeled as ever-smokers. For these individuals, we kept the cells with a matching cell type annotation with our dataset by cell type names given that the HLCA study was a main reference for our marker gene selection. A total of 33 HLCA cell types were matched to 21 of our cell types, where finer HLCA cell types were merged to match our cell type annotation (e.g., CD4 and CD8 T cells into T cell) and "nasal" cell types from HLCA were dropped. The final curated HLCA dataset comprised 136,977 cells from 27 never-smokers (10 females and 17 males) and 111,403 from 35 ever-smokers (9 females and 26 males). The numbers of the cells in each cell type displayed a similar distribution between ever- and never-smoker groups (Fig. S9).

### Lung cancer GWAS candidate causal variants (CCVs)
A total of 51 genome-wide significant loci (Supplementary Data 8) were included from 4 recent lung cancer GWAS studies: Mckay_2017 (Eur-opean population[30]), Dai_2019 (European and East Asian population[31]), Byun_2022 (meta-analysis of European, East Asian, and African population[9]), and Shi_2023 (East Asian population[32]). A total of 2574 CCVs were selected based on the following criteria (Supplemen-tary Data 9):

- For Mckay_2017, variants with log-likelihood ratio (LLR) < 1:1000 with the primary lead SNPs based on the GWAS $P$ values or $R^2 > 0.8$ with the primary lead SNPs (1000 Genomes, phase 3, EUR population).
- For Dai_2019, variants with $R^2 > 0.8$ with the primary lead SNPs (1000 Genomes, phase 3, EUR or EAS population) in 5 previously undetected loci.
- For Byun_2022, variants with LLR < 1:1000 with the primary lead SNPs based on the GWAS $P$ values or $R^2 > 0.8$ with the primary lead SNPs (1000 Genomes, phase 3, EUR, EAS, AFR, or ALL population) for 6 previously undetected loci.
- For Shi_2023, variants with LLR < 1:1000 with the primary lead SNPs based on the GWAS $P$ values or $R^2 > 0.8$ with the primary lead SNPs (1000 Genomes, phase 3, EAS population) for 13 significant loci from the discovery stage. Variants with $R^2 > 0.8$ with the pri-mary lead SNPs for 9 loci significant in replication stage were included. Three independent SNPs from conditional signals were included, and they did not have proxy variants of $R^2 > 0.8$ in EAS.

### Trait relevance score calculation
We inferred the lung-cancer-associated score for each cell using SCA-VENGE pipeline[33], based on snATAC-seq and lung cancer GWAS data. For GWAS input, we used flat probability scores evenly divided across all the CCVs in each locus. The peak-by-cell matrix of snATAC-seq data was processed using ArchR package[62]. gchromVAR was performed to cal-culate the original colocalization scores between GWAS and snATAC-seq. Seed cell proportion was set at 5%, and network propagation was applied to calculate the lung-cancer-associated score for each cell.

### Cell-type and category specificity of the accessible peaks
The specificity of the accessible peaks was assigned if they were overlapped with a single cell type or cell types in a single category (epithelial, immune, endothelial, or stromal). If a peak overlaps with cell types from multiple categories, we tested whether this peak can be mainly attributed to a single category by comparing the observed fraction of cell types to the expected 25th and 75th percentile in each category. For instance, if a peak overlaps with 7 epithelial, 2 immune, 1 stromal, and 1 immune cells, this peak is above the expected 75th percentile of epithelial (>6, considering the 8 epithelial cells in total) but not in other three categories and thus is assigned to the epithelial category. Peaks were assigned to multiple categories if they showed same levels of expected percentile in more than one category. Enrichment analysis was performed on CCV-colocalized category-specific peaks using hypergeometric test with the total number of category-specific peaks as reference sets.

### Allelic effects of predicted TF binding, cell-type-specific TF abundance assessment, and TF footprinting
Prediction of variant effects on TF binding sites was performed with the motifbreakR package[63] and a comprehensive collection of human TF-binding site models (HOCOMOCO, v11)[64]. We selected the infor-mation content algorithm and used a threshold of $10^4$ as the maximum $p$-value for a TF-binding site match in motifbreakR[63]. The allelic-binding effect was defined by the difference between alternative allele score and reference allele score larger than 0.7. "Abundant TF" in a given cell type was defined as 1) TF is expressed in > 50% of the cells in that cell type and 2) TF expression level in the same cell type is above 75th percentile of the values from all predicted TF-cell type pairs. TF footprint analysis was performed for each allelic-binding TF using the 'Footprint' function in Signac by restricting to the peak regions. A significant footprint was determined by visualizing the height of the motif-flanking region accessibility compared to the background (expected Tn5 insertion rate)[55].

### Reporter assays for variant allelic function test
Allelic transcriptional activity of two CCVs from the 15_5p15.33 locus was assessed as part of massively parallel reporter assays (MPRA)[65]. MPRA library construction, transfections, sequencing, and data ana-lyses methods were based on a previous study[37]. Briefly, MPRA libraries were generated by cloning 145 bp genomic sequence encompassing each test variant with reference or alternative allele in forward and reverse complementary direction in front of the minimal promoter of the luciferase constructs. Each test sequence was associated with 25 unique sequence tags (12 bp) at the 3′ untranslated regions of the luciferase gene. MPRA libraries were transfected to A549 lung adeno-carcinoma cell line. A linear regression was performed to assess the effect of allele on the transcript output measured by normalized tag counts across 25 different tags and 5 transfection replicates ($n = 125$ independent experiments) using the formula: Ratio = Allele + Strand (forward or reverse) + Transfection batch. To determine the sig-nificance of allelic difference, robust sandwich-type variance estimate in the Wald test was performed.

### cCRE-module, cCRE-cCRE, and cCRE-gene correlation
The co-accessible cCRE modules of two or more cCREs were identified by Cicero with Louvain community detection algorithm and co-accessible score cutoff of 0.32 (automatically defined)[24]. A more stringent co-accessible score cutoff of 0.5 was used to define "directly co-accessible" cCREs. For cCRE-gene correlation, we identified cCREs that may regulate a given gene by computing the correlation between gene expression and accessibility at nearby cCREs and correcting for bias due to GC content, overall accessibility, and peak size using Signac v1.12.0. Specifically, we performed the cCRE-gene Pearson correlation analysis by running the "LinkPeaks" function with "distance" of $10^6$ (±1 Mb of TSS), "min.cells" of 10, "$p$value_cutoff" of 0.05, and "scor-e_cutoff" of 0.05. A six-level target gene assignment for the lung cancer GWAS loci was performed based on the following criteria:

- Level 1 (module correlation with a promoter): CCV-colocalizing cCRE is a member of a cCRE module, and one or more member(s) is an annotated promoter cCRE
- Level 2 (direct correlation with a promoter): the accessibility of CCV-colocalizing cCRE is directly correlated (co-accessible score > 0.5) with the accessibility of an annotated promoter cCRE
- Level 3 (a promoter cCRE): CCV-colocalizing cCRE is annotated as a promoter
- Level 4 (module correlation with a gene expression): CCV-colocalizing cCRE is a member of a cCRE module, and the accessibility of one or more member(s) of the same module is linked to gene expression levels
- Level 5 (direct correlation with a gene expression-linked cCRE): the accessibility of CCV-colocalizing cCRE is directly correlated with that of a gene expression-linked cCRE
- Level 6 (a gene expression-linked cCRE): the accessibility of CCV-colocalizing cCRE is linked to gene expression levels

Additionally, cCRE-gene correlation analysis was performed using larger distance settings of ±2 Mb and ±5 Mb to provide insights to potential longer-range correlation (Supplementary Data 13–16).

### Reporting summary
Further information on research design is available in the Nature Portfolio Reporting Summary linked to this article.

## Data availability
All single-cell sequencing raw (fastq or bam) and pre-processed data (h5 and Seurat object) generated in this study have been deposited in the Gene Expression Omnibus (GEO) database under the accession code GSE241468. Source data are provided as a Source Data file. The processed single-cell RNAseq data for the integrated Human Lung Cell Atlas (HLCA) core dataset was downloaded from CZ CELLxGENE web portal (https://cellxgene.cziscience.com/collections/6f6d381a-7701-4781-935c-db10d30de293). Source data are provided with this paper.

## Code availability
The code for reanalyzing the data reported in this paper are deposited with detailed instructions on Github (https://github.com/pumclyy/HLISCA) and Zenodo (https://doi.org/10.5281/zenodo.13306592)[66].

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

## Acknowledgements

This work was supported by the Intramural Research Program (IRP) of the Division of Cancer Epidemiology and Genetics, National Cancer Institute (NCI), US National Institutes of Health (NIH). This work utilized the Biowulf cluster computing system at the NIH. We thank the members at the NCI Cancer Genomics Research Laboratory (CGR) and Center for Cancer Research Sequencing Facility (CCR-SF) for help with sequencing efforts and members of NHLBI flow cytometry core for help with FACS sorting. Parts of Fig. 1A and B were created with BioRender.com. This work was supported by the National Research Foundation of Korea (NRF) grant funded by the Korean government (Ministry of Science and ICT) (No. 2023R1A2C1004922; E.Y.K), National Natural Science Foundation of China (Excellent Youth Scholars Program, 32470635, 32300483, and 82090011; E.L.) and Chinese Academy of Medical Sciences Innovation Fund (2023-I2M-3-010 and 2023-I2M-2-001; E.L.), State Key Laboratory Special Fund (2060204; E.L.), State of Nebraska LB595, LB692, and NIH/NIEHS R00ES033259 awards (J.X.), Major Project of Guangzhou National Laboratory, Grant GZNL2023A02002 and GZNL2024A01003 (F.Y.), Cancer Prevention Research Interest of Texas (CPRIT) award (RR170048; C.I.A.) and National Institutes of Health (NIH) for INTEGRAL consortium (U19CA203654 and R01CA243483; C.I.A., R03CA277197; J.B.). We thank Michelle Antony and Elelta Sisay for proofreading the manuscript and Hangnoh Lee for helpful advice on the analyses. The content of this publication does not necessarily reflect the views or policies of the US Department of Health and Human Services, nor does the mention of

trade names, commercial products, or organizations imply endorsement by the US Government.

## Author contributions

E.L., E.Y.K, and J.C. conceived and designed the project. J.H.S, J.G.L., and E.Y.K. collected and processed tissue samples. J.Y., A.K., and H.P. performed experiments. E.L., Y.L., B.L., X.S., and C.W. performed the data analyses. E.L. and J.C. led the manuscript writing process. J.S. provided statistical support. All the authors (E.L., J.Y., J.H.S., Y.L., B.L., A.K., H.P., X.S., C.W., T.L., J.X., Y.H., J.B., T.Z., W.Z., M.T.L., N.R., Q.L., Y.S.C., F.Y., C.I.A., J.S., J.G.L., E.Y.K., and J.C.) discussed the results and contributed to the manuscript writing. J.G.L., E.Y.K., and J.C. supervised the project.

## Funding

## Competing interests

The authors declare no competing interests.
