## [Peer Review File · Nature Communications]

REVIEWER COMMENTS

Reviewer #1 (Remarks to the Author): Expert in cancer genetics and risk, GWAS, and CREs

By performing a joint profiling of single cell transcriptome and chromatin accessibility on normal lung tissues, the authors revealed cell-type specific lung cancer susceptibility genes through colocalizing with GWAS loci of lung cancer. The authors claim to have identified cCREs which overlapped with 68.6% of the GWAS loci, and most of them were cell type specific, especially in epithelial. Moreover, they focused on a rare epithelial cell, AT2-proliferating cell, contributed the most of GWAS loci. Then, they sought to discovered the candidate susceptibility genes by performing a multi-level cCRE-gene linkage which categorized the cCRE module into six levels, elaborating the multiple regulatory modules of candidate causal variants-colocalizing cCRE. In addition, the authors also reported the differences in the MHC-I and MHC-II pathways by the analysis between ever- and never-smokers, suggesting that smoking status-specific immune response through MHC pathways might contribute to lung cancer risk. The work is relatively complete in identifying cell-type genetic regulation associated with lung cancer susceptibility. However, they demonstrated their own findings without the replication of alternative public data or experimental validation. Below I address the major and minor issues.

Major issues

1. The authors conducted the single-cell multi-omics sequencing on the lung tissues followed with the balance of major cell groups with 6:3:1 by FACS sorting. I presume that it is subjective to balance the proportion of the sequencing data, and was the balanced data consistent with the other results in related published studies?
2. The article provides limited presentation of quality control results for single-cell sequencing data. Apart from cell count in the supplementary tables, is there consistency in changes between chromatin accessibility and the corresponding gene expression? Similarly, the activity of snATAC-seq peaks whether exhibit the consistency with TF expression?
3. Although the sample size is small in this study, the authors still compared the gene expression between ever- and never-smokers, with only one gene reaching the significance threshold. Given the gene is regulated by the CREs, I suggest the authors can also compare the ATAC peaks between ever- and never-smokers in each cell type.
4. Following the comment 3, the significant difference in MHC-II/ MHC-I communications between ever- and never-smokers should be validated among the additional datasets.
4. The authors employed multi-level linkage approach to identify cell-specific candidate susceptibility genes which only considered the regulation of genes within 1Mb, perhaps overlooking the regulation in long range, which often represents a predominant and impactful pattern of gene regulation.
5. The figures in the article offer limited information, predominantly in the form of illustrative examples. This hinders a comprehensive understanding of the favorable and robust results. For example, line 242, the overall result of TF motif enrichment did not show in any figures or tables. Besides, many results

presented in tables could be replaced with visuals, aiding in a quicker and more visually appealing understanding, such as the Manhattan plot for supplementary table S9, S10 for GWAS loci.

6. Line 145, “Most of these cCREs were located in genic or gene promoter regions, including promoter (22%), exonic (4.3%), and intronic regions (46.1%)”. The cCREs was identified snATAC-seq that was used to reveal epigenomic elements, it is confused me that over 14,000 cCREs reside in the exonic regions.

7. The authors claim that a substantial proportion CREs were cell type-specific, and revealed a rare epithelial cell, AT2-proliferating cell, contributed the most of GWAS loci (12 loci), this important finding should be validated by profound experiments.

8. Related to the above issue, to provide supporting validation for the crucial gene TERT (Figure 3E), which exhibits specificity to basal cells and contains two CREs colocalized with GWAS loci, the authors could conduct immunohistochemical staining. This would confirm the specific expression within this cellular component.

Minor issues

1. Figure 2D, there is no legend to signify the color and size representations in the bubble plot

2. Line 297, nine GWAS loci that were identified exhibited eQTLs with candidate target genes. Please provide detailed information on the SNPs and corresponding target genes in supplementary figures.

3. As I mentioned in the major criticisms section, the cell identification was relative subjective, the sub-populations of cells should be detected.

4. It is unclear how many peaks in the results exhibit cell-type and category specificity, respectively, as defined in the methods section

Reviewer #2 (Remarks to the Author): Expert in single-cell multi-omics, computational genomics, and gene regulation

This manuscripts presents an study of a large multimodal (RNA and ATAC) single cell sequencing of lung cancer and the characterization of putative susceptibility locus. The single cell data has a impressive quality and size. I am not aware of a similar multimodal data based on clinically relevant samples. The bioinformatics analysis is based on state-of-art tools and mostly well sound. An interesting (and potentially novel) analysis is provided in Fig. 4, where authors propose distinct types of cCREs linking. The manuscript is however mostly descriptive and lacks validations. Moreover, some analysis are rather inconclusive (comparison of smokers vs. non-smokers) and should be tuned down. While the data set represents an important resource, code, pre-processed data or web-resource are not provided in the manuscript.

Specific points:

The comparison of smoking status reveals no significant differences (only a single non-coding gene is detected). Authors still report a cell-cell communication analysis. Are there statistical significance associated with this? This analysis is rather descriptive and should be toned down.

I wonder if any other clinical information (cancer staging) could be used for contrast? Alternatively, authors could explore unsupervised approaches that find groups of patients by considering expression (<https://www.biorxiv.org/content/10.1101/2023.02.23.529642v1>) and cellular distribution (<https://www.biorxiv.org/content/10.1101/2022.12.16.520739v2>).

On page 10, authors describe, "many TFs predicted to display allelic binding affinity to this variant, IRF8 was abundantly expressed in dendritic cells, a cell type where the footprints of its binding motif were also detected (Figure 3H-J; Table S9)." Figure J does not show a footprint but an average footprint profile. The footprint is the ATAC-seq profile around the exact binding site. Authors should change the text or shown the base pair resolution ATAC-seq profile around the binding site.

I could not find any processed data in GEO (R or scanpy objects) as indicated in the "Data and Code Availability." I could only find raw sequences there. Sharing the data as pre-processed would provide a unique resource for the community.

The same section indicates: "Any additional information required to reanalyze the data reported in this paper is available from the lead contact upon request." Authors should deposit their code on GitHub, as this is important for reproducibility.

Reviewer #3 (Remarks to the Author): Expert in lung cancer genetics and risk

Long et al described the transcriptome and chromatin accessibility landscape of normal lung cells at the single cell level. They performed colocalization of lung cancer risk alleles and cell-type-specific regulatory elements. They identified variants associated with TERT expression, and cell-type-specific variants targeting NRG1. They also found genes in the same risk locus with different cis-regulatory elements different which were context dependent. The study is comprehensive. The results are clearly presented. The paper can provide valuable resources for future research of cell-type-specific lung cancer function to GWAS.

One major comment is how different sample size of each cell population is handled. A co-localization observed in one cell type but not in the other can be related to less power in the other cell population.

The two variants associated with TERT expression, especially rs7725218 is pretty weak.

Additional benchmarking analysis would be helpful for the single-cell multiome protocol.

Please add more description about the cell-cell communication analysis in smokers vs non-smokers. How is communication defined? Is it based on aggregated pathway expression? If so, how were the pathways defined? More details are needed in the methods. Is the MHC-II gene suppression associated with known lung cancer risk alleles in the locus?

Given this is a resource paper, is there a way to make the data more accessible?

Response to the reviewers

We thank the reviewers for their valuable comments, which have helped improve our manuscript. Below please find our point-by-point response in blue. All the changes to the manuscript are highlighted in yellow.

Reviewer #1: Expert in cancer genetics and risk, GWAS, and CREs

By performing a joint profiling of single cell transcriptome and chromatin accessibility on normal lung tissues, the authors revealed cell-type specific lung cancer susceptibility genes through colocalizing with GWAS loci of lung cancer. The authors claim to have identified cCREs which overlapped with 68.6% of the GWAS loci, and most of them were cell type specific, especially in epithelial. Moreover, they focused on a rare epithelial cell, AT2-proliferating cell, contributed the most of GWAS loci. Then, they sought to discover the candidate susceptibility genes by performing a multi-level cCRE-gene linkage which categorized the cCRE module into six levels, elaborating the multiple regulatory modules of candidate causal variants-colocalizing cCRE. In addition, the authors also reported the differences in the MHC-I and MHC-II pathways by the analysis between ever- and never-smokers, suggesting that smoking status-specific immune response through MHC pathways might contribute to lung cancer risk. The work is relatively complete in identifying cell-type genetic regulation associated with lung cancer susceptibility. However, they demonstrated their own findings without the replication of alternative public data or experimental validation. Below I address the major and minor issues.

Thank you for the summary and general comments. Please find below our detailed responses to your concerns.

Major issues

1. The authors conducted the single-cell multi-omics sequencing on the lung tissues followed with the balance of major cell groups with 6:3:1 by FACS sorting. I presume that it is subjective to balance the proportion of the sequencing data, and was the balanced data consistent with the other results in related published studies?

Thank you for the question. We chose the balancing ratio based on previous single-cell studies of normal lung tissue and our study purpose.

Travaglini *et al.* (*Nature* 2020, PMID: 33208946) used a balancing ratio (epithelial:immune:endothelial:stromal = 2:1:1:1) for the subset of samples used for FACS sorting. Similarly, Habermann *et al.* (*Science Advances* 2020, PMID: 32832598) used a similar ratio (non-immune:immune = 2:1) for a subset of their samples. Both studies aimed to reduce the proportion of the over-represented immune cell populations and to enrich the rest, especially the epithelial cells that are preferentially lost during the preparation of single-cell suspension.

Consistent with these studies, our balancing ratio was roughly ~2:1 (epithelial to immune cells) to enrich the epithelial populations, while retaining a substantial proportion of immune and the other cell groups (based on the estimation by flow cytometry profile; **Figure 1F**). Reassuringly, a median of 38.8% epithelial and 31.4% immune population per sample estimated by flow cytometry were exceedingly close to the final proportions of 40.2% epithelial and 30.8% immune cell types based on single-cell RNA expression and clustering (**Figure 2B**).

Both Travaglini *et al.* and Habermann *et al.* combined different types of balanced and unbalanced samples (mixture of blood, normal lung, and diseased lung), and therefore it is difficult to assess the direct consequences of their balancing ratios. However, they detected 17.1% and 32.6% epithelial cells in their respective final datasets, which suggested a substantial enrichment compared to typically observed pre-balancing proportions (median ~10% in our dataset). As a reference point, Travaglini *et al.* estimated the relative abundance of epithelial cells and immune cells based on the literature of classical studies as ~23% and ~20%, respectively. Our cell-type balancing was in line with these previous

studies by achieving an intended enrichment of the epithelial populations at a similar or higher proportion in the final dataset.

We edited the **Results** and **Methods** to clarify this point to the readers.

- (Line 106; Removed) (6:3:1 mixing of presumed “epithelial”, “immune”, and the remaining fractions)
- (Line 514) Balancing of fresh-dissociated lung cells using FACS sorting has been reported by other groups, applying roughly 2:1 ratio of epithelial to immune cells and achieving up to >30% of epithelial cells.
- (Line 522) To enrich for epithelial cells, which are considered to have key roles in lung cancer etiology, we collected all “epithelial” cells from EPCAM⁺CD45⁻ gates and balanced the ratios to roughly 6:3:1 (“epithelial”: “immune”: “endothelial or stromal”).

2. The article provides limited presentation of quality control results for single-cell sequencing data. Apart from cell count in the supplementary tables, is there consistency in changes between chromatin accessibility and the corresponding gene expression? Similarly, the activity of snATAC-seq peaks whether exhibit the consistency with TF expression?

Thank you for these questions. We applied three additional quality control measures to assess the consistency in changes between chromatin accessibility and gene expression data or TF expression.

First, to enable a gene-level comparison between gene expression and chromatin accessibility, we inferred the gene activity score based on the chromatin accessibility of the promoter region (2000bp upstream of the transcription start site or TSS) and gene body of each gene (Pliner *et al. Mol Cell* 2018; PMID: 30078726). Among 41 canonical cell-type marker genes, 37 showed a correlation (Pearson correlation coefficient $R > 0.5$, $P < 0.01$) between percentages of cells expressing a marker gene and those with a non-zero gene activity across the cell types. Seventeen markers showed a stronger correlation (Pearson correlation coefficient $R > 0.8$, $P < 1 \times 10^{-5}$, **Figure S2**), suggesting consistent changes between chromatin accessibility and the corresponding gene expression for the cell-type defining markers.

Second, to assess the consistency of the two modalities at the dataset level, we performed a co-embedding analysis (**Methods**, Line 604). Specifically, we imputed RNA expression from snATAC-seq data using gene activity score based on the computed anchors, followed by merging datasets of two modalities and visualizing them into a co-embedding UMAP plot. As shown in **Figure S3**, the co-embedding of cells identified from the snRNA-seq and snATAC-seq data showed substantial overlap between the two modalities.

Figure S2. Seventeen canonical markers showing high consistency between gene expression and gene activity score.

Figure S3. The visualization of UMAP co-embedding of RNA and ATAC modality.

Third, to assess the consistency between chromatin accessibility and TF expression levels, we inspected our TF abundance and footprinting data. Among 196 unique cCREs overlapping CCVs, 116 cCREs harbored TF motifs predicted to show allelic binding to a CCV. For 46 of those cCREs (40%), the expression of one or more TFs were “abundant” in one or more cell types where the cCREs were detected (“abundant” defined as expressed in > 50% of the cells with a level ranked at the top 25% in that cell type among all 218 predicted TFs for the 116 cCREs), which suggested consistent cell-type specific changes between chromatin accessibility and TF expression for this subset. Notably, TF motifs found in 30 of the 46 cCREs (65%) also showed visually detectable average footprint profile within the dataset-wide accessible chromatin regions, adding support of potential binding events. These TFs included “lineage-specific” TFs such as FLI1, IRF1, and ETS1 (“lineage-specific” defined as inducing differentiation of human pluripotent stem cells; Ng *et al. Nature Biotechnology* 2020; PMID 33257861). Another example of consistency between TF levels and chromatin accessibility is IRF8, which is predominantly expressed in dendritic cells (**Fig. 3I**), and its average footprint profiles were most strongly detected in dendritic cells (**Fig. 3J**). Although TF expression levels are not the sole determinants of the level of chromatin accessibility harboring corresponding binding motifs, these data suggested consistent changes between cell-type specific chromatin accessibility and TF expression in a subset of the tested cCREs.

We now present this data in **Results** (Line 138), **Methods** (Line 581 and 604), and **Figures S2-3**.

- (Line 138) To validate the consistency between chromatin accessibility and the corresponding gene expression, we calculated gene activity scores based on the chromatin accessibility of the promoter and gene body region (**Methods**). Among 41 canonical cell-type marker genes, 37 showed a correlation (Pearson correlation coefficient $R > 0.5$, $P < 0.01$) between the percentages of cells expressing a marker gene and those with a non-zero gene activity across the cell types, and 17 of them showed a stronger correlation (Pearson correlation coefficient $R > 0.8$, $P < 1 \times 10^{-5}$; **Figure S2**). Moreover, the co-embedding of the cells identified from snRNA-seq and snATAC-seq data showed substantial overlap between the two modalities (**Figure S3**).

3. Although the sample size is small in this study, the authors still compared the gene expression between ever- and never-smokers, with only one gene reaching the significance threshold. Given the gene is regulated by the CREs, I suggest the authors can also compare the ATAC peaks between ever- and never-smokers in each cell type.

Thank you for this important suggestion. We compared the ATAC peaks between ever- and never-smokers in each cell type by separately detecting peaks from ever- and never-smoker groups, which identified 1,677 smoking-responsive cCREs (FDR < 0.05, logistic regression; **Table S7; Methods**). Epithelial (e.g., AT2 and ciliated) and immune (NK and T) cell populations showed high numbers of smoking-responsive cCREs (**Figure S7A**). Notably, these included cCREs located in the promoter or intron of 6 of 24 suggestive smoking-responsive genes, *SYN3*, *SLC16A1*, *NKAIN2*, *NHS*, *ACTN2*, and *MCF2L* (**Table S6**). Reassuringly, all six cCREs showed differential accessibility in the same direction as the differential gene expression based on the smoking status, and three of them were located in the gene promoters (*SLC16A1*, *NKAIN2*, and *ACTN2*) and detected in the same cell types where the smoking-responsive genes were detected. We provide the full list of smoking-responsive cCREs in **Table S7** and illustrate the three representative cCREs in **Figure S7B-D**.

Figure S7. Smoking-responsive cCREs. (A) The number of significant smoking-responsive cCREs between ever- and never-smokers in each cell type. The bars are colored by cell categories (green: epithelial, red: endothelial, purple: stromal and blue: immune.) (B-D) Peak signals of representative smoking-responsive cCREs and expression levels of their target genes (B: *SLC16A1*, C: *NKAIN2* and D: *ACTN2*) in the cell types that they were identified. The sequencing tracks representing chromatin accessibility of smoking-responsive cCREs are displayed. Each track represents the aggregated snATAC signal, normalized by the total number of reads in the regions. The regions of smoking-responsive cCREs are highlighted in orange. Gene expression levels are shown on the right side of normalized signal track. At the bottom, gene tracks show the direction of transcription by arrows and exons by boxes. The boxes in peak track represent cCREs called in the genomic region.

We now present this data in **Results** (Line 184), **Methods** (Line 632), **Figure S7**, and **Tables S6-7**.

- (Line 184) Similarly, we compared the chromatin accessibility between ever- and never-smokers in each cell type to nominate “smoking-responsive cCREs” (**Methods**). A total of 1,677 unique “smoking-responsive cCREs” were identified (FDR < 0.05, logistic regression, **Table S7**; **Figure S7**), including those within the promoter or intron of suggestive smoking-responsive genes with a matching direction (**Table S6**), which corroborated the findings in gene expression.

4. Following the comment 3, the significant difference in MHC-II/ MHC-I communications between ever- and never-smokers should be validated among the additional datasets.

Thank you for raising this point. We validated the trend of MHC-II/ MHC-I communications between ever- and never-smokers in an independent dataset of 248,380 human lung cells (Sikkema *et al. Nat Med.* 2023; PMID: 37291214). Specifically, we obtained the core dataset (“normal” lung tissues) from the integrated Human Lung Cell Atlas (HLCA) portal and compiled the cells from the individuals with available smoking history information. This process resulted in a curated set of 136,977 cells from 27 never-smokers (10 females and 17 males) and 111,403 from 35 ever-smokers (9 females and 26 males) (**Figure S9A**; **Methods**). Across 21 matching cell-types, numbers of included cells showed similar distributions between never- and ever-smokers (**Figure S9B**). We performed CellChat analysis on this dataset focusing on the ligand-receptor pairs for MHC-I and MHC-II pathways. As shown in **Figure S9C**, the MHC-I pathway communications were stronger in ever-smokers compared to never-smokers across most of the tested cell types individually as well as in a combined sum across all the cell types. On the other hand, MHC-II pathway communications were stronger in never-smokers compared to ever-smokers in cell types such as dendritic, B, and macrophages, which are involved in MHC-II peptide presentation. A combined sum of MHC-II pathway communications across all the cell types was slightly stronger in never-smokers. These data suggested that the trend of opposing directions of MHC pathways communications between ever- and never-smokers was consistently observed in an independent dataset.

Figure S9. Validation of the MHC-I and MHC-II communication trend in the Human Lung Cell Atlas (HLCA) dataset. (A) The demographic information of ever- (blue) and never-smokers (red) from the HLCA core dataset that were included in the analysis. (B) The total number of cells across different cell types in ever- (blue) and never-smokers (red). (C) Heatmap of the communication strength of MHC-II and MHC-I pathways predicted by CellChat in ever- (blue) and never-smokers (red) across each cell type is shown on the left. Color indicates the log₂-transformed ratio of pathway-level communication strength of never- over ever-smokers (positive value or red: stronger in never-smokers, negative value or blue: stronger in ever-smokers). Gray indicates that the pathway strength is 0 in the cell types of smokers and never-smokers. The right part presents the communication strength of each pathway for ever- (blue) and never-smokers (red) summarized across all the cell types.

We now present this data in **Results** (Line 203), **Methods** (Line 655), and **Figure S9**.

- (Line 203) To further validate the opposing trend of MHC-I and -II pathway communications in an independent dataset, we compiled normal human lung samples from 35 ever- and 27 never-smokers (248,380 cells across 21 matching cell types; HLCA). We observed that MHC-I pathway communications were relatively higher in ever-smokers across all the cell types, while MHC-II pathway communications were higher in never-smokers mainly in macrophage, B, and dendritic cells (**Figure S9**). Although exploratory, our findings provided a descriptive summary of cell-cell communications across the lung cell types between ever- and never-smokers.

5. The authors employed multi-level linkage approach to identify cell-specific candidate susceptibility genes which only considered the regulation of genes within 1Mb, perhaps overlooking the regulation in long range, which often represents a predominant and impactful pattern of gene regulation.

Thank you for the comment. We agree that it is valuable to consider the long-range regulation as well. In addition to the main analysis using +/-1Mb window from the TSS, we further performed the cCRE-gene linkage analysis using the range of +/-2Mb and +/-5Mb, identifying 98,325 and 188,014 unique linkages, respectively. Overlaying these linked cCREs with the CCVs from the lung cancer GWAS resulted in nomination of candidate susceptibility genes in one (2Mb window) and four (5Mb window) additional GWAS loci. To enable data sharing, we included the full lists of cCRE-gene linkage with different window sizes (1Mb, 2Mb, and 5Mb) as well as their overlap with the lung cancer CCVs as a **Supplementary File 1**. We added this information in **Results** (Line 335) and **Methods** (Line 749) sections and discussed this as a potential limitation of the main analysis (Line 470).

- (Line 335) While +/-1Mb *cis*-window likely covers main gene regulatory interactions given a median size of topologically associating domain being less than 500kb, we further explored a potential cCRE-gene linkage in wider ranges (+/-2 and 5 Mb), which nominated additional candidate susceptibility genes (**Supplementary File 1**).
- (Line 470) Third, our cCRE-gene linkage mainly considered the regulation within 1Mb distance, which may potentially overlook the long-range regulation. To begin to explore this possibility we performed and

shared the full results of cCRE-gene linkage using 2Mb and 5Mb ranges as a data resource (**Supplementary File 1**)

6. The figures in the article offer limited information, predominantly in the form of illustrative examples. This hinders a comprehensive understanding of the favorable and robust results. For example, line 242, the overall result of TF motif enrichment did not show in any figures or tables. Besides, many results presented in tables could be replaced with visuals, aiding in a quicker and more visually appealing understanding, such as the Manhattan plot for supplementary table S9, S10 for GWAS loci.

Thank you for these valuable suggestions. We generated three more Supplementary Figures (**Figures S14-16**) to summarize TF analyses results and visualize the data presented in **Tables S10** and **S11**. We edited the text referencing these new figures accordingly.

First, we visualized the overall distribution across GWAS loci of the TFs with abundant cell-type expression or with a detectable footprint profile averaged among all the peaks in **Figure S14**.

Figure S14. Distribution of the allelic transcription factors (TFs) across GWAS loci that were “abundantly expressed” or with a detectable average footprint profile. An “abundantly expressed” TF (yellow) was defined if it was expressed in >50% of the cells of a given cell type and the TF expression level in that cell type is above 75 percentile of the levels among all the predicted allelic TFs. TFs with a detectable average footprint profile (blue) were defined if the accessibility measured by Tn5 insertion in their motif-flanking regions was enriched over the background levels across the peaks in the genome in one or more cell type.

- (Line 266) Fifty-six unique allelic TFs predicted for the CCVs across 20 loci were abundantly expressed in the same cell type(s) where the cCREs were detected (**Figure S14**), suggesting a condition favorable for a binding event. We further performed TF footprinting for all the allelic TFs (**Methods**), and 82 unique TFs predicted for the CCVs across 27 loci exhibited an enriched average accessibility to their motif-flanking regions, suggesting a potential binding of these TFs to cCREs in the lung cells (**Figure S14**).

Second, we visualized the central information of **Table S10** – prioritization of the CCVs displaying either a TF footprint or cell-type matching TF abundance – by displaying the total number of variants scored 3 or 4 across the GWAS loci in **Figure S15**.

Figure S15. Total number of the CCVs scored 3 or 4 in each locus by TF expression and footprints. The cCRE-colocalized CCVs with an abundantly expressed TF or a TF footprint were assigned a score of 3 (blue). The cCRE-colocalized CCVs with an abundantly expressed TF and a TF footprint were assigned a score of 4 (green).

- (Line 271) These data allowed us to score the functionality of the CCVs based on the cCRE overlap, allelic binding TF prediction, TF abundance, and footprints (Table S10). Overall, 111 cCRE-overlapping CCVs from 29 GWAS loci were supported by either an average footprint detection or a cell-type matching TF abundance (functional score = 3), and 37 CCVs from 15 loci were supported by all four categories (functional score = 4, ranging from 1-7 CCVs per locus; median = 2), providing a substantial variant prioritization (Figure S15; Table S10).

Third, we presented the central information of Table S11 - the level-6 cCRE-gene linkage - using a LocusZoom style plot (displaying genomic locations of cCREs and their linked genes together with the association P -value and Z score information) for each GWAS locus (Figure S16).

Figure S16. A summary of lung cancer susceptibility genes based on level-6 cCRE-gene linkage per locus. The eighteen loci with a level-6 linkage are shown. Each plot is divided into three panels: the top panel presents the absolute value of the cCRE-gene linkage Z score on the y-axis, with the Z score direction color-coded (yellow for positive and blue for negative correlation). The thickness of the loop refers to the $-\log_{10}(P)$ of the cCRE-gene linkage; the middle panel displays the level-6 genes (red for newly identified genes in this study, and green for previously identified genes). Arrow refers

to the transcriptional direction. The bottom panel shows the cCREs with colors representing their cell type specificity (immune, epithelial, endothelial, stromal, or multiple cell type categories). The width of cCRE is too small to visualize in scale, and thus we show cCREs in a fixed-sized square centered in their mid-point coordinate.

- (Line 370) First, most loci displayed a complex picture of potential context-specific gene regulation and multiple candidate susceptibility genes. Specifically, 9 of 18 loci displayed connections of level-6 candidate susceptibility genes with CCV-overlapping cCREs that were detected in two or more cell-type categories, while the other half displayed category-specific gene clusters connection (**Figures S16**).

7. Line 145, “Most of these cCREs were located in genic or gene promoter regions, including promoter (22%), exonic (4.3%), and intronic regions (46.1%)”. The cCREs was identified snATAC-seq that was used to reveal epigenomic elements, it is confused me that over 14,000 cCREs reside in the exonic regions.

Thank you for this comment. We modified our statement “located in” to “overlapped with” to avoid potential confusion (Lines 149). Specifically, we used the ‘annotatePeak’ function of CHIPseeker (v1.38.0) to annotate the peaks. For instance, in **Table S10**, the cCRE (chr4:163,148,204-163,149,199) is annotated as “Exon (ENST00000422287.6/92345, exon 3 of 8)”. This means that the cCRE spanning 995bp (shaded in a screenshot below) overlaps with the 3rd exon of transcript ENST00000422287.6 (NCBI Gene ID 92345). As shown in a screenshot of this region from the UCSC Genome Browser below, the peak spans the exon 3 and surrounding intronic area which overlaps with a H3K4Me1 peak (i.e., enhancer histone mark) in 7 ENCODE cell lines. Consistently, we noted that other studies using similar annotation approaches reported a comparable proportion of “exonic” cCREs. For example, Wang *et al* (*Cell Genomics* 2022, PMID: 36277849) reported that 7.2% of cCREs overlapped with the exonic regions. These details have been added into the legend of **Table S10**.

8. The authors claim that a substantial proportion CREs were cell type-specific, and revealed a rare epithelial cell, AT2-proliferating cell, contributed the most of GWAS loci (12 loci), this important finding should be validated by profound experiments.

We appreciate the reviewer’s critical comment. We addressed this point in the following three approaches.

First, we acknowledge that the statement the reviewer pointed out could be misleading. AT2-proliferating cells displayed the largest number of cell type-specific peaks among the seven rare cell types accounting for <1% of the dataset (**Figure S21C**). However, 17 CCV-colocalized cCREs from 12 loci that were detected in AT2-proliferating cells were not exclusive to AT2-proliferating cells as none of the AT2-proliferating cell-specific cCREs overlapped with CCVs (**Figure S12B**). We revised the text to clarify this point and tone down the statement.

- (Line 238) Notably, there were substantial proportions of CCVs that were assigned to a single cell type (**Figure 3C**). Among them, macrophage- and AT2 cell-specific cCREs colocalized with the largest numbers of CCVs (**Figure S12B**). Cell type specific CCVs were also detected in rarer cell types (e.g., mesothelial and dendritic), which account for less than 1% of the total cell number. Although not cell type-specific, 17 CCV-

colocalized cCREs from 12 loci were detected in AT2-proliferating cells, suggesting the contribution of transient epithelial cell types in lung cancer susceptibility (**Table S10**).

- (Removed) Notably, 17 CCV-colocalized cCRE from 12 loci were detected in AT2-proliferating cells, suggesting the role of transient epithelial cell types in lung cancer susceptibility
- (Line 34) the mention of AT2-proliferating cells was removed from **Abstract**.

Second, to further validate AT2-proliferating cells, we performed immunohistochemical (IHC) staining of normal lung tissues using antibodies against SFTPD (a marker for both AT2 and AT2-proliferating cells) and Ki-67 (one of the markers for AT2-proliferating cells). In one of our samples, a subset of SFTPD-positive cells displayed co-staining of SFTPD and Ki-67, consistent with the marker gene expression and promoter accessibility of AT2-proliferating cells from the single-cell data. We have added representative IHC images and descriptions in **Figure S6** and **Results**.

- (Line 160) To validate this finding, we utilized immunohistochemistry (IHC) and detected Ki-67 co-stained with SFTPD in a small subset of SFTPD-expressing cells in lung alveoli (**Figure S6**).

Figure S6. Detection of potential AT2-proliferating cells co-expressing SFTPD and Ki-67 in tumor-distant normal lung tissues. Representative images of immunohistochemical staining of SFTPD and Ki-67 in normal lung alveoli. **(A)** Positive SFTPD staining in suspected AT2 or AT2-proliferating cells of lung tissue from a female smoker, FS4. **(B)** Positive Ki-67 staining in a subset of SFTPD-positive cells in an adjacent section from the same tissue. Green arrows indicate likely AT2 cells, while red arrows indicate likely AT2-proliferating cells co-expressing SFTPD and Ki-67. The scale is shown in red at the bottom right-hand corner of each image. Positive antibody staining is shown in brown and nuclei staining in blue.

Third, we inspected the marker gene alignment for AT2-proliferating cells between our dataset and HLCA dataset. We observed closely matched marker gene expression profiles that distinguish AT2-proliferating cells from AT2 cells in the HLCA dataset (**Figure S5**). We also adjusted **Figure 2D** to match the format of HLCA dataset and highlight the resemblance between two datasets. The new data is presented in **Figure S5**.

Figure S5. Marker gene expression of AT2-proliferating cells from HLCA dataset. Dot plot visualizing the normalized RNA expression of selected marker genes for AT2 and AT2-proliferating cells (AT2-pro) from HLCA dataset. The color and size of each dot correspond to the scaled average expression level and fraction of expressing cells, respectively.

- (Line 155) Notably, we identified a rare cell type, AT2-proliferating cells (0.13%), which expressed an AT2 marker (*SFTPD*) as well as the markers of cell proliferation, *STMN1*, *TYMS*, *TOP2A*, *CDK1*, and *MKI67* similar to two previous studies (“AT2-proliferating” and “cycling-AT2”) (**Figure 2D**) and the HLCA annotation (**Figure S5**).

9. Related to the above issue, to provide supporting validation for the crucial gene TERT (Figure 3E), which exhibits specificity to basal cells and contains two CREs colocalized with GWAS loci, the authors could conduct immunohistochemical staining. This would confirm the specific expression within this cellular component. We appreciate the reviewer’s comment. We performed IHC on tissue samples using antibodies against a basal cell marker, KRT17, and TERT. We detected TERT expression in a subset of KRT17-positive basal cells, where ~40% of the tested samples showed a weak staining, while one sample showed a moderate staining (moderate: clearly seen in a

×20 objective lens, weak: can be seen only with a ×40 objective lens). TERT staining was not exclusive to KRT17-positive basal cells in our samples, indicating a potential expression of TERT in other lung cell types, such as lymphocytes and macrophages. Given that *TERT* is likely regulated by multiple regulatory elements including the intronic enhancer specific to basal cells presented in **Figure 3E**, we also toned down our conclusion to avoid misleading the readers. We presented representative images of new IHC results in **Figure S13** and made edits addressing the reviewer's points in **Results, Discussion, and Abstract**.

- (Line 251) IHC staining of our lung tissues samples displayed a weak staining of TERT in a subset of KRT17-positive basal cells, although TERT detection was not exclusive to basal cells (**Figure S13**)
- (Line 259) These data suggested the potential utility of our dataset in detecting cell-type contexts of gene regulation underlying lung cancer susceptibility.
- (Removed) These data suggested that lung cancer CCVs might exert their function through the basal cell specific cCRE of lung, highlighting the utility of our approach in elucidating cell-type specific gene regulation underlying lung cancer susceptibility.
- (Line 34) The mention of TERT was removed from **Abstract**.
- (Line 430) although a weak TERT co-staining was detectable in a subset of basal cells by IHC of lung tissues.

Figure S13. Expression of KRT17 and TERT in tumor-distant normal lung tissues. Representative images from immunohistochemical staining of KRT17 and TERT in the basal cells of normal lung bronchial epithelium. Red arrows indicate KRT17-positive basal cells with detectable TERT staining at a magnification of obj. x20 (**B**) or obj. x40 (**D**). The scale is shown in red at the bottom right-hand corner of each image. (**A**) Strong KRT17 expression in basal cells of lung tissue from a female smoker, FS4. (**B**) Moderate TERT expression in a subset of basal cells in an adjacent section from the same tissue. (**C**) Moderate KRT17 expression in basal cells of lung tissue from a male non-smoker, MN4. (**D**) Weak TERT expression in a subset of basal cells in an adjacent section from the same tissue. (**E**) Weak KRT17 expression in basal cells of lung tissue from a female non-smoker, FN3. (**F**) No detectable TERT expression in a subset of basal cells in an adjacent section from the same tissue. Positive antibody staining is shown in brown and nuclei staining in blue.

Minor issues

1. Figure 2D, there is no legend to signify the color and size representations in the bubble plot

Thank you for the comment. We added a description to the legend to signify the color and size representations in **Figure 2D**.

- (Line 171) The color and size of each dot correspond to the scaled average expression level and fraction of expressing cells, respectively, with the same scale and fraction defined in panel C.

2. Line 297, nine GWAS loci that were identified exhibited eQTLs with candidate target genes. Please provide detailed information on the SNPs and corresponding target genes in supplementary figures.

Thank you for the comment. We provided a summary plot of susceptibility genes based on previous eQTL studies with detailed information on the SNPs and corresponding target genes in **Figure S17**. We also noted a typo of the locus tally and corrected the number to 10 loci (counting two MHC loci separately) in the text.

- (Line 328) Ten of these 18 loci had one or more genes identified through bulk-tissue eQTL-based colocalization or transcriptome-wide association studies (TWAS) from published studies (**Figure S17**)

Figure S17. Loci with one or more level-6 genes that were identified by bulk-tissue eQTL-based colocalization or transcriptome-wide association studies (TWAS) from published studies. The loci serial IDs are in brackets after the gene names. Note that loci 16 and 17 are two MHC loci at 6p21.33 and share some target genes. Six criteria from four published studies are presented as columns (Zhu *et al*, PMID: 33909040, TWAS and colocalization; Bosse *et al*, PMID: 31696517, TWAS; Byun *et al*, PMID: 35915169, colocalization; Shi *et al*, PMID: 37236969, TWAS and colocalization). TWAS (green) and colocalization (blue) hits are color-coded. The corresponding variants prioritized by colocalization are presented on the right side of the columns. Variants prioritized by different studies are separated by semicolons.

3. As I mentioned in the major criticisms section, the cell identification was relative subjective, the sub-populations of cells should be detected.

Thank you for this suggestion. We further performed subpopulation analysis of 16 cell types with ≥ 1000 cells and identified 55 potential subpopulations (**Figure S18**). Consistent with published studies (e.g., HLCA annotation), the T-cells were divided into two subpopulations that differentially expressed known markers of CD4- and CD8-T cells, respectively (**Figure S19A**). Similarly, the fibroblast subpopulations were divided into two subpopulations that differentially expressed known markers of adventitial and alveolar fibroblasts, respectively (**Figure S19B**). This subpopulation information is provided as part of the pre-processed data (Seurat object) shared through GEO (accession: GSE241468) and can be used in investigating potential cell states and lineage in human lung in the future studies. We noted this change in **Discussion**.

- (Line 465) To begin to explore additional cell type detection, we performed sub-population analysis for the 16 cell type clusters with ≥ 1000 cells (**Methods**) and identified 55 potential subpopulations (**Figure S18**). Among them were two T-cell subpopulations differentially expressing known markers of CD4- or CD8- T cells and two fibroblast subpopulations differentially expressing known markers of adventitial or alveolar

fibroblasts (Figure S19). These subpopulations can be used as resources for future studies investigating gene regulation in more diverse lung cell types.

Figure S18. The Weighted Nearest Neighbor (WNN) dimension visualizations of subpopulation in each cell type. The visualizations of the cell subpopulations are displayed and marked with serial numbers followed by the cell type name. The X- and Y-axis represents the first and second dimension of WNN, respectively. Cell types in each panel are ordered by the category with different color scheme (immune: blue, endothelial: red, epithelial: green, and stromal: yellow).

Figure S19. Representative subpopulations expressing known marker genes. Dot plot visualizing the normalized RNA expression of known marker genes for CD4 (*CD4*, *CD40LG*, *TNFRSF25*, *CD28*, *TRAT1*, and *CTLA4*) and CD8 (*CD8A*, *CD8B*, and *TRGC2*) T-cell subpopulations (A)

and for adventitial (*MFAP5*, *SCARA5*, and *P116*) and alveolar (*SPINT2*, *LIMCH1*, and *FGFR4*) fibroblasts subpopulations (B). The color and size of each dot correspond to the scaled average expression level and fraction of expressing cells, respectively.

4. It is unclear how many peaks in the results exhibit cell-type and category specificity, respectively, as defined in the methods section.

Thank you for this comment. There are a total of 121,088 peaks (36.6%) exhibiting cell-type specificity and 258,477 (78.2%) exhibiting category specificity. We added the details in **Results**.

- (Line 146) We then surveyed the characteristics of the detected cCREs. Similar to the observations in previous snATAC-seq and multiome studies in different tissue types, a substantial proportion of the cCREs were detected only in a single cell type ($n = 121,088$; 36.6%) (Figure S4A) or in a single category ($n = 258,477$; 78%).

Reviewer #2: Expert in single-cell multi-omics, computational genomics, and gene regulation

This manuscript presents a study of a large multimodal (RNA and ATAC) single cell sequencing of lung cancer and the characterization of putative susceptibility locus. The single cell data has an impressive quality and size. I am not aware of a similar multimodal data based on clinically relevant samples. The bioinformatics analysis is based on state-of-art tools and mostly well sound. An interesting (and potentially novel) analysis is provided in Fig. 4, where authors

propose distinct types of cCREs linking. The manuscript is however mostly descriptive and lacks validations. Moreover, some analyses are rather inconclusive (comparison of smokers vs. non-smokers) and should be tuned down. While the data set represents an important resource, code, pre-processed data or web-resource are not provided in the manuscript.

Thank you for recognizing the merit of our data and providing constructive suggestions. Please find below the point-by-point responses to your concerns.

Specific points:

The comparison of smoking status reveals no significant differences (only a single non-coding gene is detected). Authors still report a cell-cell communication analysis. Are there statistical significance associated with this? This analysis is rather descriptive and should be toned down.

Thank you for the comments. We added the details of defining cell-cell communication, including the statistical significance in **Methods** and **Results**.

- (Line 644) Specifically, a significant cell-cell communication was detected at the level of ligand–receptor pairs, based on a statistical test that randomly permutes the group labels of cells (permutation $P < 0.05$) and then recalculates the communication probability. Known ligand–receptor pairs from the CellChatDB database were used to compute the communication probability at the pair level, which were summarized into communication strength at the pathway level. The differences of summarized probabilities between ever- and never-smokers was used to nominate the top-ranking differential pathways. The KEGG pathway database (<https://www.genome.jp/kegg/pathway.html>) was used to infer the relationship between ligand–receptor pairs and pathways.
- (Line 192) Based on the known ligand-receptor pairs in CellChatDB, significant cell-cell communications were detected at the pair level and then summarized at the pathway level, for the comparison between the cells from ever- and never-smokers (**Methods**).

Nevertheless, we agree that the analysis is rather descriptive because there is no statistical test applied to identify the top-ranking differential pathways between ever- and never-smokers. We have toned down our statement to account for this limitation in **Results** and **Discussion**.

- (Line 208) Although exploratory, our findings provided a descriptive summary of cell-cell communications across the lung cell types between ever- and never-smokers
- (Line 450) Our descriptive cell-cell communication analysis suggested the MHC-I and MHC-II pathways displaying an opposite direction of changes between ever- and never-smokers

I wonder if any other clinical information (cancer staging) could be used for contrast? Alternatively, authors could explore unsupervised approaches that find groups of patients by considering expression (<https://www.biorxiv.org/content/10.1101/2023.02.23.529642v1>; MOFA; now published in PMID: 37991480) and cellular distribution (<https://www.biorxiv.org/content/10.1101/2022.12.16.520739v2>; PILOT; now published in PMID: 38177382).

We thank the reviewer for the insightful suggestions. As described in **Methods**, our samples were tumor-distant (more than 2 cm away from the tumor edge) normal lung tissues mainly from stage-I lung adenocarcinoma patients (> 75% of the patients; revised **Table S1**), from which making the distinctions between cancer stages might be difficult. Instead, we explored the suggested unsupervised approaches to find data-driven groups or differential latent factors between smokers and non-smokers. As the reviewer mentioned, the first suggested unsupervised method, multi-omics factor analysis (MOFA; now published in PMID: 37991480), focused on latent expression signature, and the second one, Patient-Level distances from single cell genomics and pathomics data with Optimal Transport (PILOT; now published in PMID: 38177382), is mainly based on changes of cellular trajectories/clusters and compositions associated with

disease progression. To apply an unsupervised approach to distinguish the main groups in our dataset (by smoking status and sex), we chose to perform MOFA, as we do not expect major changes in cellular composition between these groups. After optimizing parameters, 5,000 highly variable genes based on snRNA-seq data and 25 *cis*-regulatory topics from scATAC-seq data employing *cisTopics* (Gonzalez-Blas *et al. Nat Methods* 2019, PMID: 30962623) were used in MOFA model. This process identified 20 latent factors, which explained 39.5% and 82.1% of variance by gene expression levels and scATAC-seq based topics, respectively (**Figure R1A**; note that “R” convention was used for the figures and tables provided to reviewers but not included in the manuscript). To assess whether there is a natural grouping based on latent factors, we visualized each factor and observed that all 20 factors varied among cell types (first 4 factors are shown in **Figure R1B**) and variations of each factor between cell types were larger than those between groups by smoking status and sex (Factor 1 is shown as an example in **Figure R1C**). These results indicated that latent expression signatures in normal lung tissues in our dataset were varied mainly by cell types, and distinction by sex or smoking status was not achieved with the current approach (**Figure R1B-C**). Nonetheless we agree that an unsupervised grouping of samples is an important next step to pursue with our dataset, which warrants a future study.

Figure R1. Multi-omics factor analysis (MOFA) to identify smoking- and sex-based sample grouping. (A) Two heatmaps at the top demonstrate the percentages of variance explained by each factor generated from RNA expression (left) and ATAC based topics (right). The bar plot at the bottom shows the

total percentage of variance explained by these two types of data. (B) The distributions of top 4 factors illustrate that the variations of factors were driven by cell type heterogeneities. (C) Factor 1 is shown as an example for comparison among groups by sex and smoking status within each cell type, which indicated that the main difference of Factor 1 came from cell type difference rather than the four groups.

On page 10, authors describe, "many TFs predicted to display allelic binding affinity to this variant, IRF8 was abundantly expressed in dendritic cells, a cell type where the footprints of its binding motif were also detected (Figure 3H-J; Table S9)." Figure J does not show a footprint but an average footprint profile. The footprint is the ATAC-seq profile around the exact binding site. Authors should change the text or shown the base pair resolution ATAC-seq profile around the binding site.

Thank you so much for this valuable comment. We modified our statement in the indicated part and further edited other parts of the manuscript accordingly.

- (Line 279) Among many TFs predicted to display allelic binding affinity to this variant, IRF8 was abundantly expressed in dendritic cells, a cell type where the average footprint profiles of its binding motif were also detected
- (Line 268) 82 unique TFs predicted for the CCVs across 27 loci exhibited an enriched average accessibility to their motif-flanking regions
- (Line 272) supported by either an average footprint detection or a cell-type matching TF abundance
- (Line 308) the average footprint profile of the IRF8 across all detected peaks in all cell types. The top three cell types (color-coded) showing average footprint profiles for IRF8 motif are dendritic, B, and monocytes. The remaining cell types are in gray. Footprint profiles were corrected for Tn5 insertion bias

I could not find any processed data in GEO (R or scanpy objects) as indicated in the "Data and Code Availability." I could only find raw sequences there. Sharing the data as pre-processed would provide a unique resource for the community.

Thank you for this suggestion. We initially deposited raw sequence files in GEO and provided h5 matrices including the meta-data as supplementary files. We fully agree that sharing the pre-processed data will provide a unique resource for the community. Therefore, we added a Seurat object including post-QC data with annotation information into the same GEO series (accession: GSE241468) and revised the "Data and Code Availability" section accordingly.

- (Line 753) All single-cell sequencing raw (fastq or bam) and pre-processed data (h5 and Seurat object) are deposited in the Gene Expression Omnibus (GEO) database under the accession GSE241468.

The same section indicates: "Any additional information required to reanalyze the data reported in this paper is available from the lead contact upon request." Authors should deposit their code on GitHub, as this is important for reproducibility.

Thank you for this suggestion. We have deposited the code with detailed instruction on Github, which could be accessed via the following link: <https://gitfront.io/r/liyy/Cm3wtcsNtBtJ/16-multiome>.

- (Line 754) The code for reanalyzing the data reported in this paper was deposited with detailed instruction on Github via the following link: <https://gitfront.io/r/liyy/Cm3wtcsNtBtJ/16-multiome>.

Reviewer #3: Expert in lung cancer genetics and risk

Long et al described the transcriptome and chromatin accessibility landscape of normal lung cells at the single cell level. They performed colocalization of lung cancer risk alleles and cell-type-specific regulatory elements. They identified variants associated with TERT expression, and cell-type-specific variants targeting NRG1. They also found genes in the same risk locus with different cis-regulatory elements different which were context dependent. The study

is comprehensive. The results are clearly presented. The paper can provide valuable resources for future research of cell-type-specific lung cancer function to GWAS.

Thank you for the summary and the acknowledgement of the value and quality of our study.

One major comment is how different sample size of each cell population is handled. A co-localization observed in one cell type but not in the other can be related to less power in the other cell population.

We thank the reviewer for this critical comment. As reviewer mentioned, identifying peaks requires a sufficient number of cells within the cluster, and peak calling could be influenced by cell numbers especially in rare cell types (Heumos *et al. Nat Rev Genet* 2023; PMID: 37002403). As shown in **Figure S21A** (copied below), the total numbers of peaks within cell types displayed a weak but positive correlation with cell numbers (Pearson $R^2 = 0.36$, $P = 0.0026$; dashed blue line). Notably, among the abundant cell types (proportion > 1%; 16 out of 23 cell types accounting for 97.9% of the total cell number) there is no significant correlation between cell numbers and total peak numbers (Pearson $R^2 = 0.08$, $P = 0.29$; solid blue line), indicating that the cell numbers might not affect peak calling efficiency for most of the cell types. Simultaneously, cell type specific peak numbers displayed no significant correlation with the total peak numbers (Pearson $R^2 = 0.011$, $P = 0.63$; **Figure S21C**). Similarly, a rank test between cell type specific peak numbers and the total peak numbers showed significant difference ($P = 2.38 \times 10^{-7}$, Wilcoxon signed-rank test; **Figure S21B**). These data are consistent with previous benchmarking results of a scATAC-seq protocol demonstrating that the total number of differentially accessible regions remained largely unaffected by cell numbers (De Rop *et al. Nat Biotechnol.* 2023; PMID: 37537502).

In terms of co-localized CCVs, cell types with the largest numbers of cell-type specific colocalized CCVs were macrophages, AT2, and fibroblasts but not the most abundant ones (**Figure S12B**). In addition, there were two cell-type specific co-localized CCVs in rarer cell types (mesothelial and dendritic), suggesting relative independence between the numbers of cells and co-localized CCVs. We further performed an enrichment analysis using hypergeometric test of CCV-colocalized category-specific peaks with the total number of category-specific peaks as reference sets. Of 196 peaks colocalized with 323 CCVs, 130 were category-specific (57 epithelial, 50 immune, 13 endothelial, and 10 stromal). Total numbers of category-specific peaks are 102,907 (epithelial), 71,546 (immune), 42,738 (endothelial), and 41,286 (stromal). Among four categories, immune cell-specific cCREs displayed a significant enrichment of CCV-colocalized cCREs (adjusted $P = 0.0198$; **Figure S12A**; **Table R1**; note that “R” convention was used for the figures and tables provided to reviewers but not included in the manuscript). Together these data indicated that our main conclusion about the importance of epithelial and immune cell types in lung cancer susceptibility might not be largely affected by the power differentials based on the cell number differences across the cell types. We have added these additional analyses in **Results, Methods, Figure S12** and **S21**.

- (Line 236) These category-specific CCV-colocalized cCREs were significantly enriched in immune cells (adjusted $P = 0.0198$, Hypergeometric test; **Figure S12A**), which was consistent with the observation in SCAVENGE analysis.
- (Line 571) As previously reported, peak calling could be influenced by cell numbers especially in rare cell types⁵⁶. In our dataset, the total numbers of peaks within cell types displayed a weak but positive correlation with cell numbers (Pearson $R^2 = 0.36$, $P = 0.0026$; **Figure S21A**). Notably, among the abundant cell types (proportion > 1%; 16 out of 23 cell types accounting for 97.9% of the total cell number) there is no significant correlation between cell numbers and total peak numbers (Pearson $R^2 = 0.08$, $P = 0.29$; **Figure S21A**), indicating that the cell numbers might not affect peak calling efficiency for most of the cell types. Moreover, cell type specific peak numbers displayed no significant correlation with the total peak numbers ($P = 2.38 \times 10^{-7}$, Wilcoxon signed-rank test, Pearson $R^2 = 0.011$, $P = 0.63$; **Figure S21B-C**).

Figure S21. Association between cell numbers and peak numbers across the cell types. (A) Relationships between total numbers of peaks and cell numbers within each cell type. The dashed line shows the association among all cell types, and the solid line shows association among abundant cell types (cell proportion > 1% of total cell number). (B) Upset plot visualizes the numbers of total peaks (right purple bars) and cell type-specific peaks (top green bars). (C) Relationship between numbers of total peaks and cell type-specific peaks within each cell type. Statistical significance was tested by Pearson correlation.

Figure S12. Enrichment and cell type specificity of CCV-colocalized cCREs. (A) The bar plot exhibits the number of cCREs co-localized with CCVs among total category-specific cCREs. The bars are colored by the $-\log_{10}$ transformed adjusted p values from hypergeometric test, where immune cells show a significant enrichment (adjusted P = 0.0198). (B) The number of CCVs colocalized with cell-type specific cCREs in each cell type. The bars are colored by cell categories (blue: epithelial, purple: immune, green: endothelial, and yellow: stromal).

Table R1

Category	# Category-specific peak	# CCV-colocalized category-specific peak	Category-specific peak ratio	Colocalized peak ratio	P-value	Adjusted P-value	q-value
Epi	102907	57	102,907/258,477	57/130	0.19720519	0.39441039	0.31137662
Imm	71546	50	71,546/258,477	50/130	0.00495541	0.01982164	0.01564867
Endo	42738	13	42,738/258,477	13/130	0.98754261	0.99832603	0.78815213
Stroma	41286	10	41,286/258,477	10/130	0.99832603	0.99832603	0.78815213

The two variants associated with TERT expression, especially rs7725218 is pretty weak.

Thank you for this comment. We agree that the allelic effect of the two variants, especially rs7725218, in reporter gene expression is not strong. We toned down the statement accordingly in **Results** and **Discussion** and removed the mention of *TERT* in **Abstract**.

- (Line 256) The results demonstrated that one variant displayed significant allelic transcriptional activity at false discovery rate (FDR) < 1% with the lung cancer risk-associated allele showing higher levels, albeit with a modest effect size (rs7726159, \log_2 fold change = 0.18, FDR = 1.76×10^{-4}), while the other was not significant (rs7725218, \log_2 fold change = -0.05, FDR = 0.043) (**Figure 3F**). These data suggested the potential utility of our dataset in detecting cell-type contexts of gene regulation underlying lung cancer susceptibility.
- (Line 429): “Due to the small number of cells in this population of cells, cCRE-gene linkage was not observed for *TERT* expression, although a weak TERT co-staining was detectable in a subset of basal cells by IHC of lung tissues. Notably, the risk allele of one of the colocalizing variants displayed higher transcriptional activity

in A549 lung cancer cells, although the effect size was modest. It is conceivable that multiple other context-dependent CREs could influence *TERT* expression in lung tissues.

- (Removed) Overall, our findings suggested that the lung cancer associated variants may contribute to *TERT* expression.
- (Line 34): The mention of *TERT* was removed from **Abstract**.

Additional benchmarking analysis would be helpful for the single-cell multiome protocol. Thank you for this valuable suggestion. We addressed this point in two approaches.

First, to enable a gene-level comparison between gene expression and chromatin accessibility, we inferred the gene activity score based on the chromatin accessibility of the promoter region (2000bp upstream of the transcription start site or TSS) and gene body of each gene (Pliner *et al. Mol Cell* 2018; PMID: 30078726). Among 41 canonical cell-type marker genes, 37 showed a correlation (Pearson correlation coefficient $R > 0.5$, $P < 0.01$) between percentages of cells expressing a marker gene and those with a corresponding gene activity across the cell types. Seventeen markers showed a stronger correlation (Pearson correlation coefficient $R > 0.8$, $P < 1 \times 10^{-5}$, **Figure S2**), suggesting consistent changes between chromatin accessibility and the corresponding gene expression for the cell-type defining markers.

Second, to assess the consistency of the two modalities at the dataset level, we performed a co-embedding analysis (**Methods**, Line 604). Specifically, we imputed RNA expression from snATAC-seq data using gene activity score based on the computed anchors, followed by merging datasets of two modalities and visualizing them into a co-embedding UMAP plot. As shown in **Figure S3**, the co-embedding of cells identified from the snRNA-seq and snATAC-seq data showed substantial overlap between the two modalities.

Figure S2. Seventeen canonical markers showing high consistency between gene expression and gene activity score.

Figure S3. The visualization of UMAP co-embedding of RNA and ATAC modality.

Please add more description about the cell-cell communication analysis in smokers vs non-smokers. How is communication defined? Is it based on aggregated pathway expression? If so, how were the pathways defined? More details are needed in the methods. Is the MHC-II gene suppression associated with known lung cancer risk alleles in the locus?

Thank you for the comments.

First, we added more details of cell-cell communication analysis **Methods** and **Results**.

- (Line 644) Specifically, a significant cell-cell communication was detected at the level of ligand–receptor pairs, based on a statistical test that randomly permutes the group labels of cells (permutation $P < 0.05$) and then recalculates the communication probability. Known ligand–receptor pairs from the CellChatDB database were used to compute the communication probability at the pair level, which were summarized into communication strength at the pathway level. The differences of summarized probabilities between ever- and never-smokers was used to nominate the top-ranking differential pathways. The KEGG pathway database (<https://www.genome.jp/kegg/pathway.html>) was used to infer the relationship between ligand–receptor pairs and pathways.
- (Line 192) Based on the known ligand-receptor pairs in CellChatDB, significant cell-cell communications were detected at the pair level and then summarized at the pathway level, for the comparison between the cells from ever- and never-smokers (**Methods**).

Second, we did observe cases where the MHC-II gene suppression is associated with known lung cancer risk alleles in the GWAS locus. For example, based on the GTEx portal (<https://gtexportal.org/home/>), two cCRE-colocalized variants, rs140272475 and rs114774714, in the locus 17_6p21.33 are linked to MHC class II genes (*HLA-DQB1* and *HLA-DRB5*) in our multi-level cCRE-gene linkage. The lung cancer risk alleles of both variants (rs140272475-A and rs114774714-T) were significantly associated with the decreased expression of *HLA-DQB1* and *HLA-DRB5* in the GTEx lung eQTL dataset. However, these were anecdotal examples, and there are other eQTL connections in the opposite directions as well. Further studies will be needed to understand the molecular underpinning of the MHC pathways in the context of lung cancer susceptibility. We toned down our statements about the conclusion and interpretation of our findings in **Results** and **Discussion**.

- (Line 208) Although exploratory, our findings provided a descriptive summary of cell-cell communications across the lung cell types between ever- and never-smokers
- (Line 450) Our descriptive cell-cell communication analysis suggested the MHC-I and MHC-II pathways displaying an opposite direction of changes between ever- and never-smokers
- (Line 458) Further studies will be needed to begin to understand the mechanistic connections between potential changes of MHC pathways in smokers and lung cancer development.

Given this is a resource paper, is there a way to make the data more accessible?

Thank you for this suggestion.

First, we added a Seurat object including post-QC data with annotation information into our GEO series which already have raw sequencing data as well as h5 matrix and meta data (accession: GSE241468).

- (Line 753) All single-cell sequencing raw (fastq or bam) and pre-processed data (h5 and Seurat object) are deposited in the Gene Expression Omnibus (GEO) database under the accession GSE241468.

Second, we have deposited the code with detailed instruction on Github, which could be accessed via the following link: <https://gitfront.io/r/liyy/Cm3wtcsNtBtJ/16-multiome>.

- (Line 754) The code for reanalyzing the data reported in this paper was deposited with detailed instruction on Github via the following link: <https://gitfront.io/r/liyy/Cm3wtcsNtBtJ/16-multiome>.

REVIEWERS' COMMENTS

Reviewer #1 (Remarks to the Author):

I was satisfied with the responses and recommended to accept.

Reviewer #2 (Remarks to the Author):

Authors have considered all my comments (data and code accessibility, adaptation of the cell-cell communication analysis and additional experiments) and changed the manuscript accordingly. I have no further comments.

Reviewer #3 (Remarks to the Author):

The authors sufficiently addressed all my comments. The paper is much improved. The study is comprehensive and solid.